# Temporal regularities shape perceptual decisions and striatal dopamine signals

Matthias Fritsche ®[1] ✉, Antara Majumdar[1], Lauren Strickland[1,2], Samuel Liebana Garcia[1], Rafal Bogacz ®[3] & Armin Lak ®[1] ✉

Perceptual decisions should depend on sensory evidence. However, such decisions are also influenced by past choices and outcomes. These choice history biases may reflect advantageous strategies to exploit temporal regularities of natural environments. However, it is unclear whether and how observers can adapt their choice history biases to different temporal regularities, to exploit the multitude of temporal correlations that exist in nature. Here, we show that male mice adapt their perceptual choice history biases to different temporal regularities of visual stimuli. This adaptation was slow, evolving over hundreds of trials across several days. It occurred alongside a fast non-adaptive choice history bias, limited to a few trials. Both fast and slow trial history effects are well captured by a normative reinforcement learning algorithm with multi-trial belief states, comprising both current trial sensory and previous trial memory states. We demonstrate that dorsal striatal dopamine tracks predictions of the model and behavior, suggesting that striatal dopamine reports reward predictions associated with adaptive choice history biases. Our results reveal the adaptive nature of perceptual choice history biases and shed light on their underlying computational principles and neural correlates.

Accurate perceptual decision-making should rely on currently available sensory evidence. However, perceptual decisions are also influenced by factors beyond current sensory evidence, such as past choices and outcomes. These choice history biases are ubiquitous across species and sensory modalities[1–14]. While maladaptive in standard randomized psychophysical experiments, choice history biases could be advantageous in natural environments that exhibit temporal regularities[15]. Crucially, however, natural environments exhibit a multitude of different temporal regularities. For instance, a traffic light that recently turned green can be expected to remain green for a while, allowing a driver to maintain speed while passing a junction. Conversely, a yellow traffic light can rapidly change to red, thus prompting a driver to decelerate. The exploitation of these various temporal regularities therefore necessitates adaptation of choices to such sequential patterns. However, the behavioral

signatures, computational principles, and neural mechanisms underlying such adaptations remain unclear.

Previous studies have demonstrated that humans and rats can adapt their perceptual choice history biases to different temporal regularities[16–18]. While mice exhibit flexible visual decision-making[3,19–23], it is not known whether they can adapt their choice history biases to temporal regularities of the environment. Moreover, the neural mechanisms underlying such adaptive perceptual choice history biases remain unknown. Midbrain dopamine neurons, and the corresponding dopamine release in the striatum, play key roles in learning[24–26]. Dopamine signals have been shown to shape the tendency to repeat previously rewarded choices, both in perceptual and value-based decision tasks[21,27–30]. Yet, the role of striatal dopamine signals in the adaptation to temporal regularities during perceptual decision-making remains unknown.

[1]Department of Physiology, Anatomy & Genetics, University of Oxford, Oxford, UK. [2]Institute of Behavioral Neuroscience, University College London, London, UK. [3]MRC Brain Network Dynamics Unit, University of Oxford, Oxford, UK. ✉e-mail: Matthias.Fritsche@dpag.ox.ac.uk; Armin.Lak@dpag.ox.ac.uk

We trained mice in visual decision-making tasks involving different trial-by-trial temporal regularities, with stimuli likely repeating, alternating, or varying randomly across trials. We show that mice can adapt their perceptual choice history biases to these different temporal regularities to facilitate successful visually-guided decisions. This adaptation was slow, evolving over hundreds of trials across several days. It occurred alongside a fast non-adaptive choice history bias, which was limited to a few trials and not influenced by temporal regularities. We show that these fast and slow trial history effects are well captured by a normative reinforcement learning algorithm with multi-trial belief states, comprising both current trial sensory and previous trial memory states. We subsequently demonstrate signatures of this learning in mice that are naive to the manipulation of temporal regularities, suggesting that this type of learning is a general phenomenon occurring in perceptual decision-making. Finally, we establish that dopamine release in the dorsal striatum follows predictions of the reinforcement learning model, exhibiting key signatures of learning guided by multi-trial belief states. Together, our results demonstrate the adaptive nature of perceptual choice history biases as well as their neural correlates and cast these biases as the result of a continual learning process to facilitate decision-making under uncertainty.

## Results

### Mice adapt perceptual choice history bias to temporal regularities

We trained male mice ($n = 10$) in a visual decision-making task (Fig. 1a). In each trial, we presented a grating patch on the left or right side of a computer screen and mice indicated the grating location by steering a wheel with their forepaws, receiving water reward for correct responses. After mice reached expert proficiency on randomized stimulus sequences, we systematically manipulated the trial-by-trial transition probabilities between successive stimuli across different days (Fig. 1b). In addition to neutral stimulus sequences in which stimulus location was chosen at random [p("Repeat") = 0.5], we exposed mice to a repeating environment in which stimulus locations were likely repeated across successive trials [p("Repeat") = 0.8], and an alternating environment in which stimulus locations likely switched from the previous trial [p("Repeat") = 0.2]. Consequently, in the repeating and alternating environments, the location of the current stimulus was partially predictable given the knowledge of the previous trial. Mice successfully mastered the task, exhibiting high sensitivity to visual stimuli (Fig. 1c; choice accuracies—neutral: 77.91% ± 0.70; alternating: 79.31% ± 0.75; repeating: 79.73% ± 0.74, mean ± SEM). In order to examine whether mice's decisions were influenced by the trial

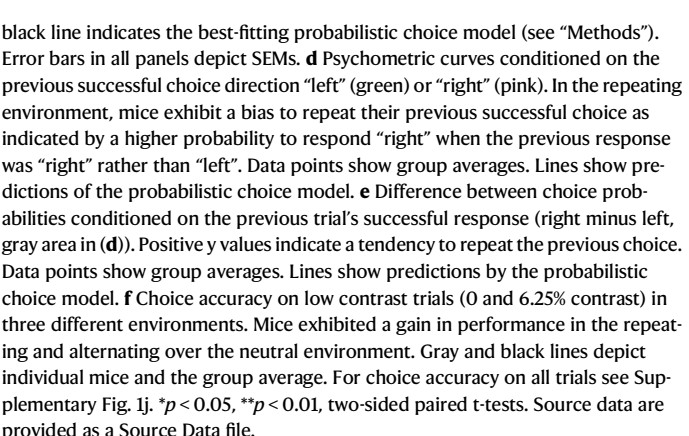

**Fig. 1 | Mice bias visual decisions to exploit temporal regularities of stimulus sequences. a** Schematic of the two-alternative visual decision-making task. Head-fixed mice reported the location (left/right) of gratings with varying contrasts by steering a wheel with their forepaws, receiving water reward for correct responses. Adapted from ref. 21. https://creativecommons.org/licenses/by/4.0/. **b** Stimulus sequences of left and right grating presentations followed distinct transition probabilities (left column), interleaved across different days. In the neutral environment, stimulus location was determined randomly (top). In the repeating and alternating environments, stimulus locations were likely repeated (middle) or alternated (bottom) across successive trials. Right column shows example sequences in each environment. Different shades of green and pink denote different stimulus contrasts, varying randomly across trials. **c** Mice exhibit expert performance, demonstrated by steep psychometric curves with near-perfect performance for easy (high contrast) stimuli (data pooled across environments). Negative and positive contrasts denote stimuli on the left and right sides, and the y-axis denotes the probability of a rightward choice. Black data points show the group average and gray lines indicate individual mice ($n = 10$ in all panels). The black line indicates the best-fitting probabilistic choice model (see "Methods"). Error bars in all panels depict SEMs. **d** Psychometric curves conditioned on the previous successful choice direction "left" (green) or "right" (pink). In the repeating environment, mice exhibit a bias to repeat their previous successful choice as indicated by a higher probability to respond "right" when the previous response was "right" rather than "left". Data points show group averages. Lines show predictions of the probabilistic choice model. **e** Difference between choice probabilities conditioned on the previous trial's successful response (right minus left, gray area in (**d**)). Positive y values indicate a tendency to repeat the previous choice. Data points show group averages. Lines show predictions by the probabilistic choice model. **f** Choice accuracy on low contrast trials (0 and 6.25% contrast) in three different environments. Mice exhibited a gain in performance in the repeating and alternating over the neutral environment. Gray and black lines depict individual mice and the group average. For choice accuracy on all trials see Supplementary Fig. 1j. *$p < 0.05$, **$p < 0.01$, two-sided paired t-tests. Source data are provided as a Source Data file.

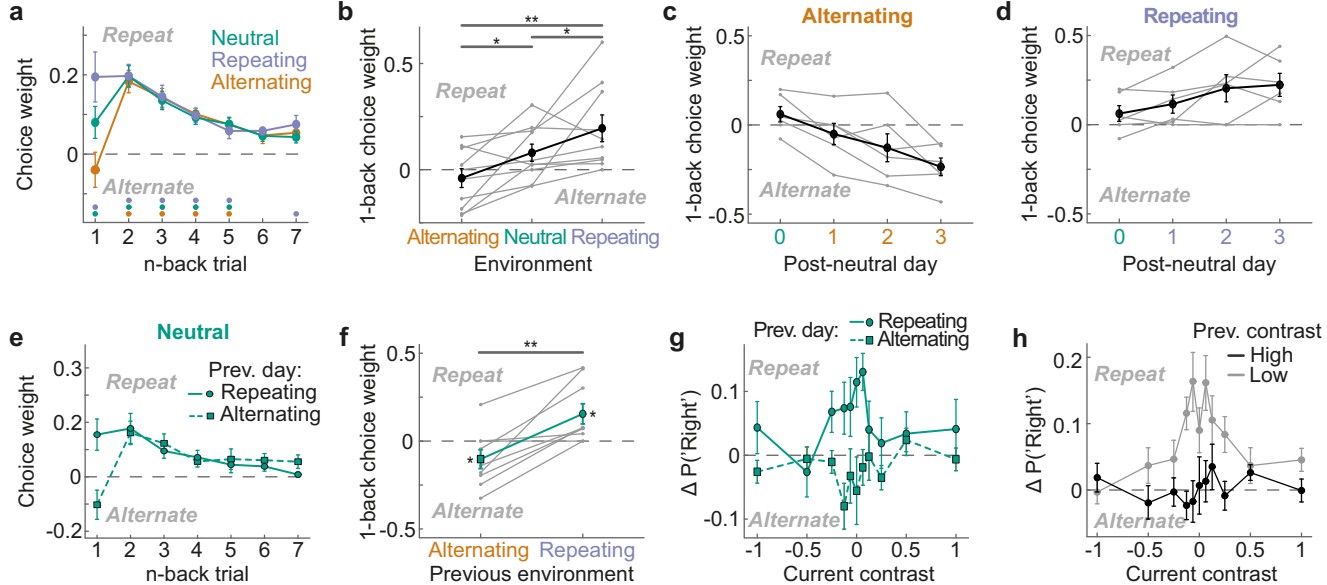

**Fig. 2 | Mice adapt their previous choice weight to different temporal regularities across multiple days. a** History kernels comprising the past seven successful choice weights of the probabilistic choice model ("Methods"; see Supplementary Fig. 1 for the full set of regression weights and parameter recovery analysis). While mice are biased by several past choices, only the previous (1-back) choice weight differs across environments. Error bars in all panels depict SEMs. Dots parallel to x-axis indicate weights significantly different from baseline, two-sided permutation test based on shuffled trial history, Bonferroni-corrected, $p < 0.007$. Sample size was $n = 10$ mice in panels a, b and h. **b** 1-back successful choice weight across environments for each mouse (gray lines) and group average (black). One-sided t-tests; repeating vs alternating: $t(9) = 3.11$, $p = 0.006$; repeating vs neutral: $t(9) = 1.97$, $p = 0.04$; alternating vs neutral: $t(9) = -2.76$, $p = 0.01$. **c** 1-back successful choice weights estimated on the first, second, and third day of alternating sessions following a neutral session ($n = 6$ mice). **d** Same as in (c), but for repeating sessions ($n =$

6 mice). **e** Choice history kernels for neutral sessions conditioned on the temporal regularity experienced on the preceding day (solid/circle: repeating; dashed/square: alternating; $n = 9$ mice in panels e, f and g). **f** 1-back successful choice weight of neutral sessions preceded by repeating (circle) or alternating (square) sessions in each mouse (gray lines) and across the population (green line). Stars denote results of one and two-sided t-tests (see main text). **g** Difference in choice probabilities conditioned on the previous trial's successful choice in neutral sessions preceded by repeating (solid/circle) or alternating (dashed/square) sessions. The differential impact of the previous regularity is most pronounced when current contrast is low. **h** Difference in choice probabilities conditioned on the previous trial's successful response split according to whether the previous trial's stimulus contrast was high (black) or low (gray). Mice are more likely to repeat the previous choice when it was based on a low rather than high contrast stimulus. $*p < 0.05$, $**p < 0.01$. Source data are provided as a Source Data file.

history, we conditioned current choices on the previous trial's successful choice direction (Fig. 1d). In the neutral environment, mice showed a subtle but consistent tendency to repeat the previous choice ($t(9) = 2.31$, $p = 0.046$, two-sided t-test), in line with previous studies[20,21]. Importantly, this choice repetition bias was increased in the repeating environment and decreased in the alternating environment, appropriate to exploit the temporal regularities of stimuli (Fig. 1e; $\Delta P("Right")$−Repeating vs. Neutral: $t(9) = 2.89$, $p = 0.018$; Alternating vs. Neutral: $t(9) = -3.65$, $p = 0.005$, two-sided paired $t$-tests). Furthermore, the influence of the previous choice was most pronounced when the current stimulus contrast was low, suggesting that mice particularly relied on learned predictions when they were perceptually uncertain. Importantly, exploiting the predictability in the repeating and alternating environments enabled mice to increase their choice accuracy relative to the neutral environment, in which stimuli were not predictable (Fig. 1f; $\Delta$Accuracy on the most difficult trials [0 and 6.25% contrast]−Repeating vs. Neutral: $t(9) = 5.28$, $p = 0.0005$; Alternating vs. Neutral: $t(9) = 2.58$, $p = 0.03$, two-sided paired $t$-tests; see Supplementary Fig. 1j for all trials). These findings indicate that mice adapt their reliance on the previous choice to the temporal regularity of the stimulus sequence, thereby improving their perceptual decisions.

## Adaptation of history bias develops over multiple days and is limited to the previous trial

To further quantify choice history biases beyond the previous trial, we fit a probabilistic choice regression model with history kernels to choices in each environment (see "Methods" and Supplementary Fig. 1 for details and parameter recovery analysis). The history kernels associated with the past seven successful choices confirmed that mice

adapted the weight of the previous (i.e., 1-back) choice to different temporal regularities (Fig. 2a and b). In contrast, the influence of choices made more than one trial ago (2- to 7-back) did not differ across environments, but steadily decayed from an initial attraction by the 2-back choice towards zero for choices made further in the past, generally ceasing to be significantly different from baseline after 5 trials (Fig. 2a; two-sided permutation tests, Bonferroni-corrected for multiple comparisons). Surprisingly, the relatively small 1-back choice weight in the neutral environment entailed that mice were more likely to repeat their 2-back choice compared to the more recent 1-back choice when acting on random stimulus sequences (Fig. 2a, green line; $t(9) = -4.08$, $p = 0.003$, two-sided t-test of 1- vs 2-back choice weights; see also Supplementary Fig. 2 for individual mice). In addition to the probabilistic choice regression model, we also confirmed this phenomenon using a model-free analysis (Supplementary Fig. 2c, $t(9) = -3.49$, $p = 0.007$, two-sided t-test of model-free 1- vs 2-back choice repetition probability). We will seek to explain this phenomenon with normative learning principles below.

In contrast to past successful choices, we found that mice tended to repeat past incorrect choices largely irrespective of the environment statistic and the temporal lag, pointing towards a long-term repetition of errors (Supplementary Fig. 1d). We hypothesized that this repetition of errors was due to prolonged periods of task disengagement in which mice largely ignored visual stimuli and instead repeatedly performed the same choice. We investigated this hypothesis by identifying engaged and disengaged trials using a modeling framework based on hidden Markov Models[31] (HMM, see "Methods" and Supplementary Fig. 3a−f). When applying the choice history analysis separately to engaged and disengaged trials, we indeed found that

mice repeated the previous incorrect choice when they were disengaged but tended to alternate after errors in the engaged state. Consistent with increased repetition of incorrect choices in the disengaged state, mice became more likely to repeat their previous choice when they committed several errors in sequence, indicative of episodes of task disengagement (Supplementary Fig. 3g). These findings support the hypothesis that long-term choice repetition after errors is strongly driven by task disengagement. Due to the low number of "engaged" error trials in the task (14% ± 0.1 of all trials, mean ± SEM), we focused on successful choice history kernels in the remainder of our analyses.

Overall, our findings indicate that although mice's choice history biases extend over several past trials, mice only adapt the influence of the previous successful choice to the temporal regularities of the stimulus sequence.

We next sought to investigate how rapidly mice adapted their previous choice weight to temporal regularities. We fit the probabilistic choice model separately to the first, second, and third day of alternating or repeating sessions following a neutral session. Mice slowly and gradually shifted their 1-back weight across days, increasingly alternating or repeating their previous choice with each day in the alternating and repeating environments, respectively (Fig. 2c and d; F(2,8) = 4.89, $p$ = 0.04, repeated-measures ANOVA). To further corroborate this slow adaptation, we analyzed neutral sessions that were preceded by either a repeating or alternating session (Fig. 2e). Consistent with slow adaptation, mice continued to weigh their previous choice according to the temporal regularity they experienced on the previous day, despite the current stimulus sequences being random (Fig. 2f). That is, mice were biased to repeat their previous choice in a neutral session preceded by a repeating session (t(8) = 2.69, $p$ = 0.01, one-sided t-test) and biased to alternate their previous choice in a neutral session preceded by an alternating session (t(8) = −1.93, $p$ = 0.045, one-sided t-test; post-repeating vs. post-alternating: t(8) = 4.18; $p$ = 0.003, two-sided paired t-test). Moreover, mice most strongly followed the regularity of the previous day when the current contrast was low (Fig. 2g), suggesting that mice integrate current sensory evidence with a flexible, but slowly acquired prediction based on past experience. Unlike the 1-back choice weight, the 2-back choice weight did not depend on previous exposure to temporal regularities (Supplementary Fig. 4). Lastly, choice repetition was also modulated by the previous trial's stimulus contrast, being more pronounced after successful choices based on low- rather than high contrast stimuli (t(9) = 4.38, $p$ = 0.002, two-sided paired t-test; Fig. 2h), similar to previous studies[9,21,32]. This modulation by past stimulus contrasts gradually decayed over n-back trials (Supplementary Fig. 5). We will seek to explain this phenomenon with learning principles below.

In summary, the results show that mice's visual decisions are biased towards the recent choice history—a bias that decays over the past seven trials. In contrast to this fast-biasing effect of the most recent choices, mice slowly adapted their 1-back choice weight to temporal regularities of the stimulus sequence over the course of hundreds of trials. Finally, even in the neutral environment mice exhibited a conspicuous reliance on their 1-back choice, repeating it less than the temporally distant 2-back choice.

**Multi-trial reinforcement learning explains choice history biases**

We next asked whether our findings could be explained by a common underlying computational principle. It has been proposed that even well-trained perceptual decision-makers exhibit choice history biases due to continual updating of choice values[21,32]. In this framework, an agent performs the visual decision-making task by combining its belief about the current stimulus (perception) with stored values for perception-choice pairs, which can be formalized as a partially observable Markov decision process (POMDP[33,34]; Fig. 3a; for a detailed description see "Methods"). In brief, on a given trial the agent estimates the probabilities $P_L$ and $P_R$, denoting the probabilistic belief that

the stimulus is on the left or right side of the screen (Fig. 3e, dark blue). These estimates are stochastic: they vary across trials even if these trials involve the same stimulus contrast. The agent then multiplies these probabilities with stored values $q_{choice,perception}$ that describe the average previously obtained reward when making a certain choice (left/right) upon observing a particular perceptual state (left/right stimulus). This yields expected values $Q_L$ and $Q_R$, describing the expected reward for either choice:

$$Q_L = \sum_{i \in \{L,R\}} P_i \cdot q_{L,P_i} \text{ and } Q_R = \sum_{i \in \{L,R\}} P_i \cdot q_{R,P_i} \tag{1}$$

The agent probabilistically chooses based on these expected values and a softmax decision rule. Following the choice $C$ (left or right), the agent observes the outcome $r$ and computes a prediction error $\delta$ by comparing the outcome to the expected value of the chosen option, $Q_C$: $\delta = r − Q_C$. This prediction error is then used to update the values associated with the chosen option $q_{C,P_L}$ and $q_{C,P_R}$ by weighing the prediction error with a learning rate $\alpha$ and the belief $P_L$ and $P_R$:

$$q_{C,P_L} \leftarrow q_{C,P_L} + \alpha \cdot P_L \cdot \delta \text{ and } q_{C,P_R} \leftarrow q_{C,P_R} + \alpha \cdot P_R \cdot \delta \tag{2}$$

The above agent has four free parameters: a sensory noise parameter, governing the variability of stimulus estimates $P_L$ and $P_R$, decision noise (softmax temperature), as well as learning rates for positive and negative prediction errors ($\alpha^+$ and $\alpha^-$). We refer to this agent as the *single-trial* POMDP RL agent, for making a choice the agent only considers its belief about the current trial's stimulus (perception, $P_L$ and $P_R$) and the associated stored perception-choice values $q_{choice,perception}$. This agent exhibits several notable features[21,32]. First, due to the trial-by-trial updating of perception-choice values, it learns the visual decision-making task from scratch (Supplementary Fig. 6a). Second, the trial-by-trial updating of perception-choice values introduces history dependencies, biasing the agent to repeat recently rewarded choices (Fig. 3b). Finally, the agent recapitulates the dependence of the choice bias on the difficulty of the previous decision: The agent is most likely to repeat a previous successful choice when it was based on a low contrast stimulus, associated with low decision confidence (Fig. 2h; Supplementary Fig. 6b). Crucially however, the agent does not explain choice history biases across different temporal regularities. In particular, when confronted with neutral (random) stimulus sequences, it produces a monotonically decaying history kernel, instead of the observed increase of choice weights from 1- to 2-back (Fig. 3b, green). Furthermore, in the alternating environment, the agent fails to capture the alternation tendency in the 1-back choice weight. (Fig. 3b, orange). Due to this failure to adapt to the alternating temporal regularity, the model underestimates the mice's behavioral choice accuracy in the alternating environment (Supplementary Fig. 6c). Finally when transitioning from a repeating or alternating environment into the neutral environment, the agent exhibits no carryover of previously acquired history dependencies, unlike the substantial carryover seen in the empirical data (Fig. 3c).

Having established that the single-trial POMDP RL agent is unable to account for the mice's adaptation to temporal regularities, we considered a simple extension to this model. In particular, we assumed that when determining the value of the current choice options, the agent not only considers its belief about the current stimulus (perception, $P_L$, and $P_R$) and the associated perception-choice values (Fig. 3d and e, dark blue), but additionally relies on its memory of the previous trial's successful choice (Fig. 3d and e, pink; $M_L$ and $M_R$). That is, analogous to computing choice values from a belief about the current stimulus, the agent combines its memory of the previous trial's rewarded choice with a separate set of memory-choice values

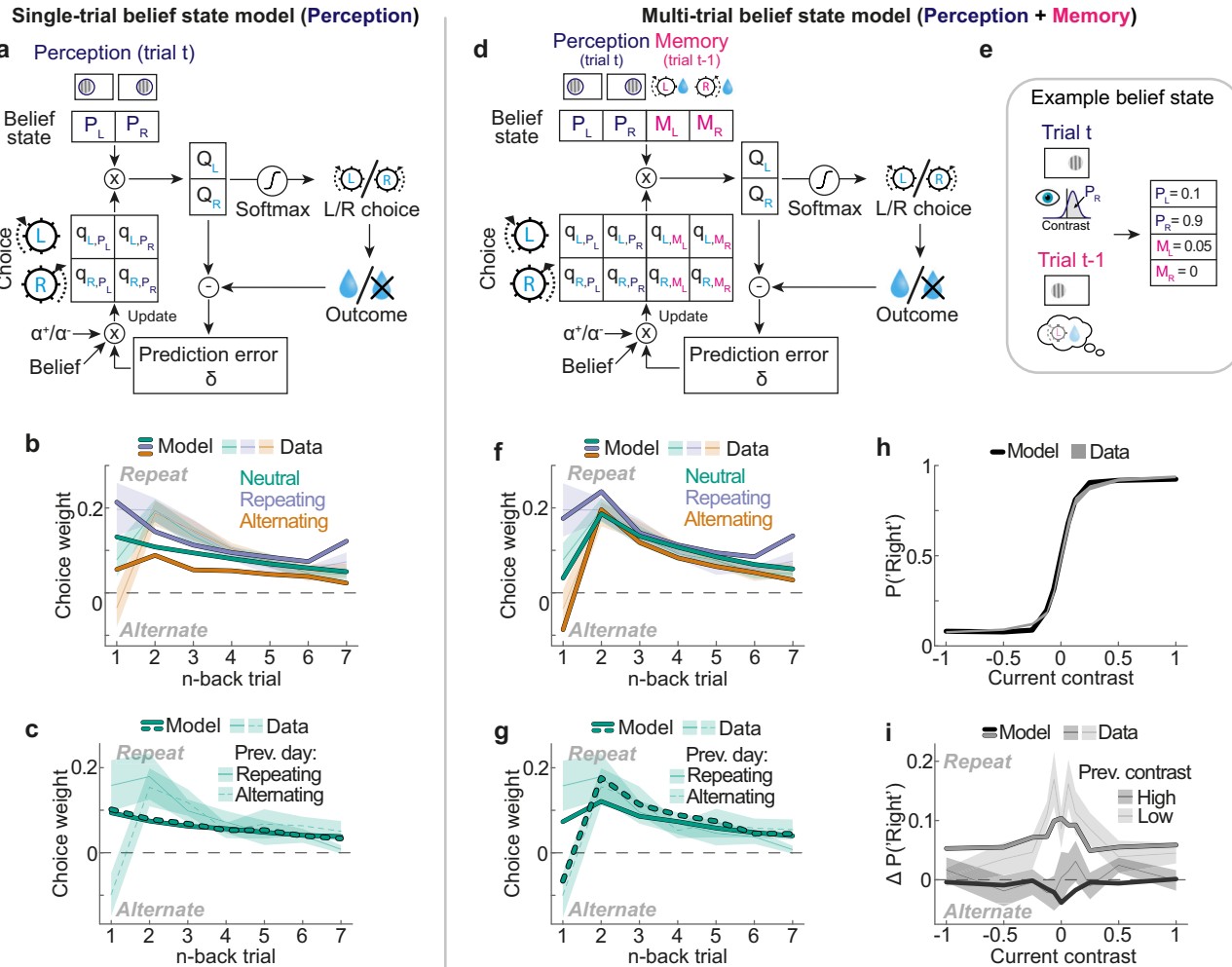

**Fig. 3 | A normative reinforcement learning algorithm with multi-trial belief states captures choice history biases across different environments. a** A normative model of decision-making and learning. The single-trial belief state model performs the task by combining its belief about the location of the current visual stimulus (top; dark blue) with stored perception-choice values. The model iteratively updates these values by means of a weighted prediction error (see "Methods"). **b** Choice history kernels of the best fitting single-trial belief state model. Iterative updating of perception-choice values leads to choice repetition, but the model cannot produce a 1- to 2-back increase in neutral choice weights (green), nor the 1-back choice alternation in the alternating environment (orange). Shaded lines and regions in all panels depict empirical means ± SEMs. **c** Choice history kernels in neutral sessions following a repeating (solid line) or alternating session (dashed line). Unlike mice, the single-trial model does not exhibit a carryover of 1-back choice weights adapted to the previous regularity. **d** The multi-trial belief state model not only considers its belief about the current visual stimulus (top; dark blue) but additionally relies on its memory of the previous trial's successful choice (top; pink), together with a separate set of

memory-choice values. **e** Example belief state. The agent has a strong belief that the current stimulus is on the right side and remembers that the previous trial's rewarded choice was left. Eye symbol used with permission from the Twemoji project. https://creativecommons.org/licenses/by/4.0/ **f** Choice history kernels of the best fitting multi-trial belief state model. The model captures the characteristic 1- to 2-back increase in neutral choice weights (green), and the 1-back choice alternation in the alternating environment (orange). **g** Choice history kernels in neutral sessions following a repeating (solid line) or alternating session (dashed line). The multi-trial model exhibits a carryover of adapted choice history weights. **h** Psychometric curves of the mice (grey), and the multi-trial model (black). The model accurately captures the mice's dependence of choice (y-axis) on current contrast (x-axis). **i** Difference in choice probabilities conditioned on the previous trial's successful response, split according to whether the previous trial's stimulus contrast was high (black) or low (gray). The multi-trial model (lines) captures the mice's increased tendency to repeat the previous choice when it was based on a low rather than high contrast stimulus (black and gray shaded regions). Source data are provided as a Source Data file.

($q_{choice,memory}$; Fig. 3d, pink). These memory-choice values describe the expected reward of a particular current choice (left/right) depending on the rewarded choice of the previous trial. The agent thus computes the expected reward for current left and right choice options, $Q_L$ and $Q_R$, as the sum of perception-based and memory-based reward expectations:

$$Q_L = \sum_{i \in \{L,R\}} P_i \cdot q_{L,P_i} + \sum_{j \in \{L,R\}} M_j \cdot q_{L,M_j} \text{ and}$$
$$Q_R = \sum_{i \in \{L,R\}} P_i \cdot q_{R,P_i} + \sum_{j \in \{L,R\}} M_j \cdot q_{R,M_j} \tag{3}$$

where $P$ and $M$ represent perceptual and memory belief states, respectively. Following the choice and outcome, the agent updates the perception-choice and memory-choice values associated with the selected choice, using the same learning rate $\alpha$:

$$q_{C,P_L} \leftarrow q_{C,P_L} + \alpha \cdot P_L \cdot \delta \text{ and } q_{C,P_R} \leftarrow q_{C,P_R} + \alpha \cdot P_R \cdot \delta \tag{4}$$

$$q_{C,M_L} \leftarrow q_{C,M_L} + \alpha \cdot M_L \cdot \delta \text{ and } q_{C,M_R} \leftarrow q_{C,M_R} + \alpha \cdot M_R \cdot \delta \tag{5}$$

We refer to this agent as the *multi-trial* POMDP RL agent, as it considers both its belief about the current trial's visual stimulus

(perception, $P_L$ and $P_R$) and its memory of the previous trial's successful choice ($M_L$ and $M_R$) when making a choice in the current trial. Compared to the single-trial agent, the multi-trial agent has only one additional parameter (memory strength), controlling how strongly the agent relies on its memory of the previous choice for current decisions. Similar to the single-trial agent, the multi-trial agent captured the mice's dependence on the current and previous stimulus contrasts (Fig. 3h and i). Strikingly, however, the agent was also able to capture the pattern of choice history biases across different temporal regularities. First, for random stimulus sequences, the agent produced the distinctive decrease in 1- compared to 2-back choice weights (Fig. 3f, green). Second, the agent accurately captured the mice's tendency to alternate the previous choice in the alternating environment (Fig. 3f, orange), and repeat the previous choice in the repeating environment (Fig. 3f, blue), while maintaining similar 2- to 7-back choice weights. Due to its ability to adapt to the alternating regularity, the agent successfully captured the mice's higher empirical choice accuracy in the alternating compared to the neutral environment (Supplementary Fig. 6c). Finally, the agent exhibited a substantial carryover of adapted 1-back choice weights when transitioning from the repeating or alternating into the neutral environment (Fig. 3g). Accordingly, the multi-trial POMDP RL model provided a significantly better fit to the mice's choice data than the single-trial model (F(1,41) = 12.63, $p = 0.001$, F-test; ΔBIC = 8.52). Extending the multi-trial POMDP RL model with exponentially decaying memory, not limited to the 1-back trial, did not further improve the model fit (F(1,40) = 0.37, $p = 0.55$, F-test; ΔBIC = −3.41; see "Methods" and Supplementary Fig. 7). Importantly, the fit of the multi-trial model was achieved by fitting a single set of parameters to the data of all three temporal regularities, suggesting that the empirical differences in choice history biases arose from a fixed set of learning rules that created different choice dynamics depending on the regularity of the input sequence.

In order to better understand how the multi-trial agent was able to capture our findings, we inspected the trajectories of perception-choice and memory-choice values across the different environments. We found that perception-choice values underwent strong trial-by-trial fluctuations, but remained overall stable across different temporal regularities (Supplementary Fig. 8a). In contrast, memory-choice values changed slowly over the course of hundreds of trials, and diverged in the different environments (Supplementary Fig. 8b and c). The slow change in memory-choice values was driven by a subtle reliance on memory, relative to perception, when deciding about the current choice, thereby leading to small updates of memory-choice values. Notably, the updating of perception-choice values is relatively rigid, promoting a tendency to repeat successful choices regardless of the temporal regularity of the environment. Conversely, memory-choice values can grow flexibly to either facilitate or counteract the repetition tendency (Supplementary Fig. 8c). Since memory only comprised the previous trial's choice, this facilitating or counteracting effect was limited to the 1-back choice weight. Importantly, in the neutral environment, any history dependency attenuates task performance. In this environment, the multi-trial agent used its memory to counteract the 1-back repetition bias introduced by the updating of perception-choice values, leading to a decreased 1- relative to 2-back choice weight. The reliance on memory thus allowed the agent to become more neutral in its reliance on the 1-back choice, thereby increasing task performance. Finally, the slow trajectory of memory choice values offers an explanation for why mice did not develop a pronounced 1-back repetition bias in the repeating environment (Fig. 3f, blue). Both neutral and alternating environments discourage the model from repeating the 1-back choice, promoting memory-choice values in favor of alternations. Since repeating sessions were interleaved with neutral and alternating sessions, the model therefore was not given enough time to adapt its memory-choice values to produce strong 1-back repetition biases during repeating sessions,

resulting in a muted repetition bias, similar to the empirically observed pattern in mice.

Together, our results demonstrate that mice's choice history biases and their slow adaptation to different temporal regularities can be explained by a normative reinforcement learning algorithm with multi-trial belief states, comprising both current trial sensory and previous trial memory states.

## Mice naive to temporal regularities exhibit signatures of multi-trial learning

Mice exhibited a key signature of the multi-trial POMDP RL agent, displaying a decreased tendency to repeat the 1- relative to 2-back choice when acting on completely random stimulus sequences. We wondered whether this reliance on memory was driven by successful learning of different temporal regularities (Fig. 2a), or whether it is a general phenomenon observed in animals that did not experience such temporal regularities. To investigate this question, we analyzed publicly available choice data of 99 naive mice, which had not experienced repeating or alternating regularities, and were trained to expertize with random stimulus sequences in an experimental setup similar to ours[20] (Fig. 4a–c). Similar to the mice of the current study, mice more strongly repeated their previous successful choice when the previous contrast was low rather than high (Fig. 4b; t(98) = 10.17, $p < 2.2e-16$, two-sided paired t-test), which is an important feature of confidence-weighted updating of choice values (see Fig. 3i). Crucially, while mice were biased to repeat successful choices of the recent past, they also exhibited a reduced 1- compared to 2-back choice repetition bias, replicating this signature of multi-trial learning in mice naive to temporal regularities (Fig. 4c; t(98) = −3.47, $p = 0.0008$, two-sided paired t-test). Importantly, our analysis was restricted to the neutral sessions of the International Brain Laboratory's (IBL) dataset, i.e., before blocks with different stimulus probabilities were introduced (see "Methods" for details). Conversely, in the IBL's main task, in which stimuli were more frequently presented on one side in blocks of 20 to 100 trials, mice exhibited a monotonically decreasing choice history kernel with a larger 1- compared to 2-back weight (Supplementary Fig. 9). This is likely driven by the high stimulus repetition probability within each block lasting for dozens of trials, strongly encouraging mice to repeat the previous choice[35].

Overall, our analysis shows that even without exposure to biased temporal regularities, mice exhibit a key signature of multi-trial learning, suggesting that learning based on multi-trial belief states is a general strategy in visual decision-making of mice.

## Reduction in previous choice weight is not due to sensory adaptation

The decreased tendency to repeat the 1- relative to 2-back choice and the increased probability to repeat decisions based on low sensory evidence are key signatures of the multi-trial POMDP RL agent. However, both phenomena could also be the signature of spatially-specific sensory adaptation. It is well known that the visual system adapts to visual input, typically leading to reduced neural responses for repeated or prolonged stimulus presentations[36–44], and inducing repulsive biases in behavior[45–49]. Neural adaptation to high contrast is believed to reduce perceptual sensitivity to subsequently presented low contrast stimuli[50,51]. If high contrast stimuli indeed reduce the perceptual sensitivity to subsequent stimuli, this may explain the reduced bias to report a grating presented at the same location as the previous grating (i.e., repeating the previous successful choice), and would entail a particularly strong reduction following gratings with the highest contrast. Together with a monotonically decaying choice repetition bias (Fig. 4d, blue), a short-lived sensory adaptation bias (Fig. 4d, red) may thus mimic both signatures supporting multi-trial reinforcement learning (Fig. 4d, striped). In order to investigate this possibility, we exploited a crucial necessary condition of the adaptation hypothesis,

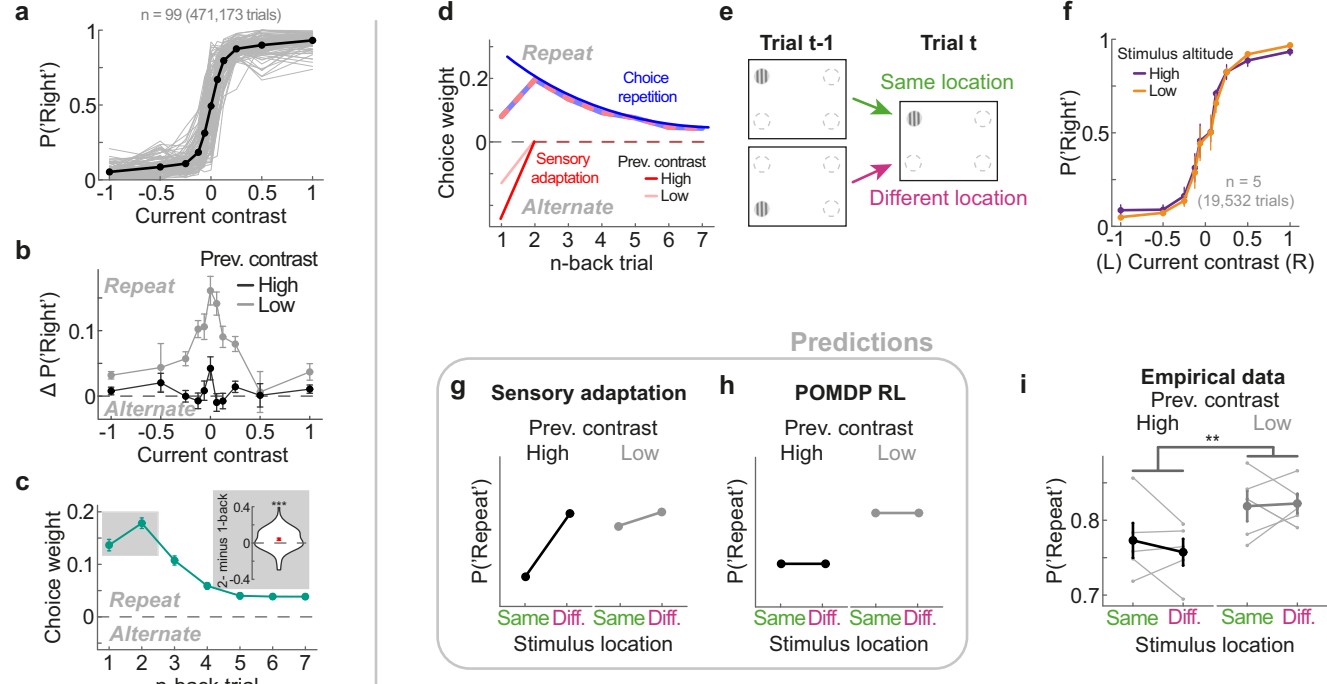

**Fig. 4 | Naive mice exhibit a signature of multi-trial learning, which is not explained by spatially-specific sensory adaptation. a** Mice performed a visual decision-making task, similar to the one of the current study[20] (*n* = 99, (**a–c**)). We exclusively analyzed sessions in which mice had mastered the task with random stimulus sequences. Mice showed high sensitivity to the current stimulus contrast. Gray lines show individual mice, whereas black data points show the group average. See also Fig. 1c. **b** Difference in choice probabilities conditioned on the previous trial's successful response, split according to whether the previous trial's stimulus contrast was high (black) or low (gray). Mice are more likely to repeat the previous choice when it was based on a low rather than high contrast stimulus. In all empirical panels (**b**, **c**, **f**, and **i**) data points show the group average and error bars depict SEMs. **c** History kernel comprising the past seven successful choice weights of the probabilistic choice model ("Methods"). While naive mice generally tended to repeat their most recent choices, they exhibited a reduced 1- relative to 2-back choice weight (see inset, two-sided paired t-test, t(98) = −3.47, *p* = 0.0008), which is a key signature of our multi-trial POMDP RL model. **d** Schematic of how a monotonically decaying choice repetition bias (blue) together with a short-lived sensory adaptation bias (red) could lead to the empirically observed choice history kernel (striped), explaining both the 1- to 2-back increase in choice weights and the modulation of choice weights by previous contrast. **e** Schematic of the altered visual decision-making task. Stimuli were presented at one of four spatial locations, unlike the main task with two stimulus locations. Mice had to report whether the

stimulus was on the left or right side of the screen. Successive stimuli could thus be presented at the same (green arrow) or different spatial location (pink arrow), even when those stimuli required the same choice (here left). **f** Mice (*n* = 5) exhibit expert-level task performance. We expressed the probability of a rightward decision (y-axis) as a function of the signed stimulus contrast (x-axis). Positive contrasts denote stimuli on the right side. Mice showed a high sensitivity to visual stimuli, regardless of whether stimuli were presented low (yellow) or high (purple) in their visual field. **g** Predictions of the spatially-specific sensory adaptation hypothesis. Mice should be less likely to repeat the previous successful choice (y-axis) when successive stimuli are presented at the same spatial location (x-axis, green), rather than at the different spatial location in the same hemifield (x-axis, pink). This effect should be particularly pronounced when the previous stimulus had high contrast (left subpanel), serving as a potent adapter. **h** Predictions of the POMDP RL model with confidence-weighted value updates. Mice should be more likely to repeat a previously successful choice when it was based on a low rather than high contrast stimulus (left vs right subpanels), but this effect should not vary with changes in spatial location (x-axis, green vs pink). **i** The mice's (*n* = 5) choice repetition probabilities are in line with the predictions of the POMDP RL model, and inconsistent with spatially-specific sensory adaptation. Grey thin lines depict individual mice. Repeated-measures ANOVA. **\*\****p* < 0.01, **\*\*\****p* < 0.001. Source data are provided as a Source Data file.

namely that sensory adaptation needs to be spatially-specific, reducing perceptual sensitivity for subsequent stimuli presented at the same location as the previous stimulus, but less strongly when the stimulus location changes. To test this, we performed a new experiment in which we presented small gratings in one of the four corners of the screen. Mice reported whether the stimulus was on the left or right side of the screen, regardless of its altitude. This experiment thereby allowed us to manipulate whether successive stimuli were presented at the same or different spatial location (lower and upper visual field), even when those stimuli required the same choice (left or right; Fig. 4e). Crucially, the POMDP RL framework, which posits confidence-dependent updating of choice values, predicts an effect of previous contrast, with an increased probability to repeat successful choices based on low sensory evidence (see Fig. 3h), but no influence of whether the previous and current stimuli were presented at the same or different spatial locations (Fig. 4h). Conversely, spatially-specific sensory adaptation predicts that the probability to repeat the previous

successful choice is reduced when previous and current stimuli are presented at the same location, and relatively increased when the location changes, due to a release from adaptation—an effect that should be particularly pronounced when the previous contrast was high (Fig. 4g). Mice (*n* = 5) successfully reported the horizontal location of the current stimulus (left or right), both when stimuli were presented at a low or high vertical location (Fig. 4f). Furthermore, we verified that mice did not make substantial eye movements towards the visual stimuli (Supplementary Fig. 10). Crucially, mice showed an increased tendency to repeat their previous successful choice when the previous contrast was low (F(1,4) = 30.06, *p* = 0.005), but no effect of a change in stimulus altitude (F(1,4) = 0.24, *p* = 0.65) and no interaction between changes in stimulus altitude and contrast (F(1,4,) = 0.51, *p* = 0.51, repeated-measures ANOVA). A Bayes factor analysis revealed moderate evidence against the hypothesis that a change in spatial location from a previous high contrast stimulus leads to an increased repetition bias (BF10 = 0.25), a central prediction of the

sensory adaptation hypothesis. Our results are thus inconsistent with sensory adaptation, and point towards confidence-dependent updating of choice values underlying choice repetition. Furthermore, the decreased tendency to repeat the 1- relative to 2-back choice could not be explained by mice pursuing two distinct decision-making strategies on distinct sets of trials, either alternating the previous choice while acting largely independent of the longer-term history or repeating past choices monotonically weighted by their n-back position. Instead, the 1- to 2-back increase in choice weight was pervasive, whenever mice were engaged with the decision-making task (Supplementary Fig. 11).

### Striatal dopamine tracks behavioral choice history biases

Finally, we sought to elucidate the neural bases of adaptive choice history biases. Central to the hypothesis of reinforcement learning underlying choice history biases is that mice compute reward predictions and reward prediction errors while making perceptual decisions. The activity of midbrain dopamine neurons and the resulting dopamine release in the striatum are strongly implicated in this process[21,52–55]. A key target area implicated in learning of stimulus-choice associations is the dorsolateral striatum[56] (DLS). We measured dopamine release in the DLS, using ultra-fast dopamine sensors[57] (GRAB$_{DA2m}$) in combination with fiber photometry (Fig. 5a), in order to compare striatal dopamine signals with our multi-trial POMDP RL model.

We measured dopamine release in the DLS while mice ($n = 6$) performed our visual decision-making task (Fig. 1a). Since we found

signatures of adaptation to trial history even in the neutral (random) environment (see Fig. 4a–c), we focused on measuring choice behavior and dopamine release using random stimulus sequences, maximizing the number of trials in this condition ($n = 11,931$ trials). Mice successfully mastered the decision-making task (Fig. 5b) and exhibited a similar choice history kernel to previous experiments (Fig. 5f; c.f. Figs. 2a and 4c). Importantly, they expressed the characteristic increase in choice weights from the 1- to 2-back trial (Fig. 5f, inset; t(5) = -2.73, $p = 0.02$, one-sided t-test), which is a key distinguishing feature between the multi- and single-trial models. Dopamine release in the DLS was strongly modulated both at the time of stimulus and outcome (Fig. 5c–e and Supplementary Fig. 12f). Following the stimulus presentation, dopamine increased with stimulus contrast (F(1.58,7.92) = 16.995, $p = 0.002$, repeated-measures ANOVA; Fig. 5d and e), largely independent of the stimulus side relative to the recorded hemisphere (F(1,5) = 0.69, $p = 0.44$, repeated-measures ANOVA; Supplementary Fig. 12b). Conversely, following reward delivery dopamine negatively scaled with stimulus contrast (F(1.6,8) = 117.34, p = 1.8 × 10⁻⁶, repeated-measures ANOVA), yielding the highest dopamine release for rewarded zero contrast trials (Supplementary Fig. 12f and g). These signals are consistent with dopamine encoding the expected reward value during stimulus processing, for which a high contrast stimulus predicts a highly certain reward (model Q; Fig. 5g, black line), and dopamine encoding the reward prediction error during outcome (model δ), for which the maximal surprise occurs when receiving a reward given a maximally uncertain stimulus. Further

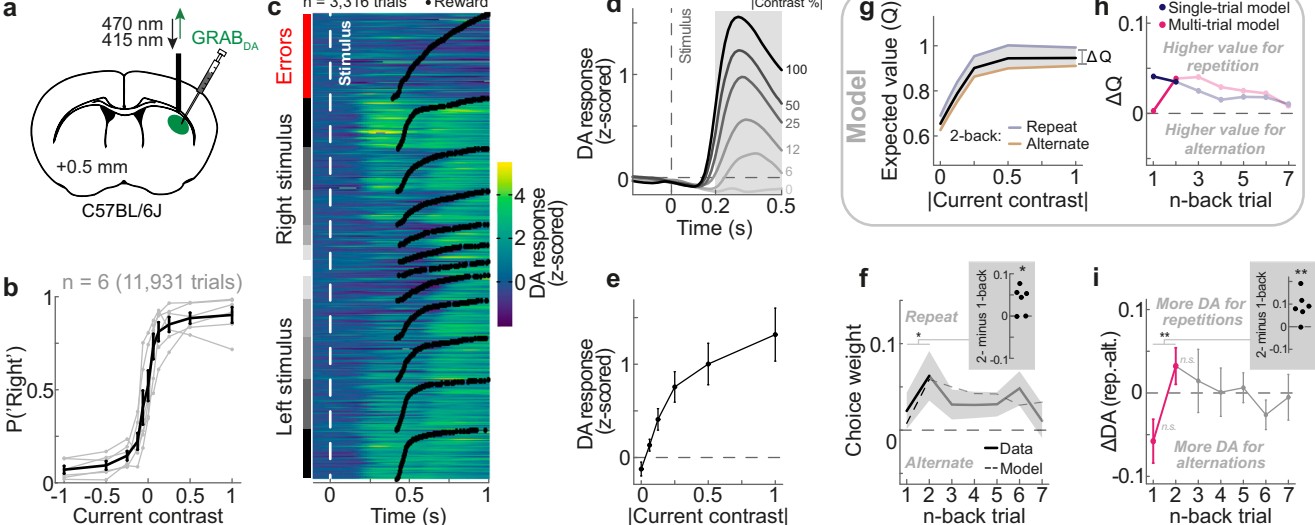

**Fig. 5 | Striatal dopamine tracks the expected reward value of the multi-trial POMDP RL model. a** Schematic of fiber photometry in the dorsolateral striatum (DLS), imaging dopamine release using ultra-fast dopamine sensors (GRAB$_{DA2m}$). **b** Psychometric curves of mice *(n = 6)* during the dopamine recording experiment. Gray lines show individual mice, whereas black data points show the group average. Error bars in all panels depict SEMs. **c** Trial-by-trial dopamine responses from all sessions of an example animal, aligned to stimulus onset (white dashed line) and sorted by trial type (left column) and outcome time (black dots). **d** Group-average dopamine response (*n* = 6 mice, **d**, **e**, **f**, and **i**), aligned to stimulus onset (gray dashed line), split by stimulus contrast (gray to black; correct trials only). Gray shaded area indicates the stimulus time period over which we averaged stimulus responses (**e** and **i**; excluding time points after reward delivery). **e** Average stimulus-evoked dopamine responses as a function of current absolute contrast (rewarded trials only; averaged over gray shaded area in (**d**)). **f** History kernel of the probabilistic choice model fit to mice data (solid line) and the predicted history kernel of the multi-trial POMDP RL model (dashed line). Mice exhibit a higher 2- compared to 1-back choice weight (inset, one-sided t-test, t(5) = −2.73, $p = 0.02$). Shaded region depicts SEMs. **g** Expected reward value Q (black) of the multi-trial POMDP RL model

as a function of current contrast (absolute value, i.e., independent of its L or R position), separately when the current stimulus is on the same (repeat, blue) or opposite side (alternate, orange) as the 2-back stimulus (current and previous rewarded trials only). *Q* reflects the expected value before the choice, computed by summing $Q_L$ and $Q_R$ weighted by the probability of making a left and right choice. For $Q_C$, the expected value after the choice, see Supplementary Fig. 13. **h** Difference in *Q* between repetitions and alternations of stimulus side (ΔQ) as a function of n-back trial (current and previous rewarded trials only). The single-trial (blue) and multi-trial models (pink) make opposite predictions about the difference between 1- and 2-back trials. While the single-trial model predicts a higher ΔDA for the 1-back versus to 2-back trial, the multi-trial model predicts a higher ΔDA for the 2- compared to 1-back trial. **i** Difference in stimulus-evoked dopamine responses between repetitions and alternations of stimulus side (ΔDA) as a function of n-back trial (current and previous rewarded trials only). Mice exhibit a higher 2- compared to 1-back ΔDA (inset, two-sided t-test, t(5) = 3.51, $p = 0.017$). 1- and 2-back ΔDA are not significantly different from zero (n.s., two-sided t-tests), respectively. Source data are provided as a Source Data file.

evidence supporting the hypothesis that dopamine responses during the stimulus period reflected the expected reward value $Q$ comes from the observation that dopamine scaled with the uncertainty of the previous stimulus, consistent with the model predictions (Supplementary Fig. 12c and d).

In order to examine effects of trial history on dopamine responses, we focused on dopamine release during the stimulus period, which unlike reward-related responses were not complicated by an overlap of stimulus and reward responses caused by $GRAB_{DA}$ sensor dynamics (Supplementary Fig. 12f). An important feature of the POMDP RL model is that the expected reward value Q not only depends on the contrast of the current stimulus, but also on the history of past choices and outcomes. In particular, the expected reward is higher if the current stimulus is presented on the same rather than the opposite side as previously rewarded trials (Fig. 5g, blue vs orange), promoting the behavioral choice repetition bias. Crucially however, since the multi-trial agent uses its memory of the previous rewarded choice to reduce the 1-back repetition bias in the neutral environment, the difference in expected reward between stimulus repetitions and alternations ($\Delta Q$) is larger for the 2- compared to 1-back trial, mimicking the empirically observed choice history kernel (Fig. 5h, pink; c.f. Fig. 5f). This is in contrast to the single-trial agent, which predicts a larger $\Delta Q$ for the 1- compared to 2-back trial (Fig. 5h, blue). To test whether stimulus-evoked dopamine tracked the multi-trial agent's dynamic of $\Delta Q$ across trials, we analogously computed the difference in stimulus-evoked dopamine between stimulus repetitions and alternations of the n-back rewarded side ($\Delta DA$, Fig. 5i). We found that dopamine responses indeed tracked the multi-trial model's predictions of $\Delta Q$: the dopamine response to the current stimulus was larger when the current stimulus was a repetition of the 2-back compared to the 1-back trial ($\Delta DA$, 2- minus 1-back: $t(5) = 3.51$, $p = 0.017$, two-sided t-test; $p = 0.018$, two-sided permutation test based on shuffled trial history), and gradually decayed across further n-back trials (Fig. 5i). Importantly, this dependence of dopamine on stimulus repetition or alternation of the 1- or 2-back trial was not evident during the pre-stimulus period of the current trial ($\Delta DA$, 2- minus 1-back: $t(5) = -0.30$, $p = 0.78$, two-sided t-test; Supplementary Fig. 12e), and thus was not a carryover of residual dopamine from previous trials. Given the fast response times of mice and sensor dynamics, it was not possible to clearly separate dopamine signals before and after the current choice. However, the pattern of $\Delta Q$ (Fig. 5g and h) holds regardless of whether it is calculated before or after the choice (Supplementary Fig. 13), thus making similar predictions for pre-outcome dopamine signals. We note that while the multi-trial agent exhibits a near-zero, but slightly positive 1-back $\Delta Q$, we observed a numerically negative 1-back $\Delta DA$, which was not statistically significantly different from zero ($t(5) = -2.20$, $p = 0.08$, two-sided t-test). We speculate that such a negative 1-back $\Delta DA$ in the DLS could be driven by an unequal weighting of reward predictions, calculated based on the perceptual and memory components of the multi-trial belief state. Indeed, reward expectations based solely on memory exhibit a negative 1-back $\Delta DA$, and we found that an overweighting of this memory-based expectation could approximate the empirically observed DA release (Supplementary Fig. 14). Although speculative, we therefore consider it possible that DLS DA release might report a reward expectation that is slightly skewed towards memory-based expectations. It is possible that other striatal regions such as DMS, which receives more input from visual cortical areas[58,59], might more strongly encode reward expectations based on perception. More experiments will be necessary to investigate this hypothesis.

Together, our results indicate that pre-outcome dopamine in the DLS closely tracks the expected reward value of the multi-trial POMDP RL agent. Given dopamine's prominent role in striatal neural plasticity and learning, we speculate that it may thus play a role in mediating adaptive choice history biases in perceptual decisions.

## Discussion

Our world presents a multitude of temporal regularities, allowing observers to predict the future from the past. Here, we show that mice can exploit such regularities to improve their perceptual decisions by flexibly adapting their reliance on past choices to the temporal structure of the stimulus sequence. We find that this adaptation of perceptual choice history biases is well captured by a normative reinforcement learning algorithm with multi-trial belief states, comprising both current trial sensory and previous trial memory states. Moreover, we show that learning guided by multi-trial belief states occurs even in mice that never experienced the manipulation of temporal regularities, suggesting that multi-trial learning may be a default strategy when making perceptually uncertain decisions. Lastly, we demonstrate that dopamine release in the DLS closely tracks behavioral biases and reward predictions in the multi-trial reinforcement learning model, pointing towards a plausible teaching signal linked to the learning and exploitation of temporal regularities in perceptual decisions.

It has been previously proposed that perceptual choice history biases can be explained by reinforcement learning mechanisms that are continually engaged to adjust perceptual decisions, even in highly trained decision-makers[32]. In the POMDP RL framework, observers continually evaluate and update the values of different choice options given their sensory confidence, choice, and feedback[33,34,52]. This framework has been fruitful in explaining the emergence of choice history biases, their dependence on previous sensory confidence, and the adaptation of choices to changes in reward value[21]. However, it does not explain how mice adapt their choice history biases to different temporal regularities, and in particular how mice learn to alternate from a previously rewarded choice when stimulus sequences favor alternations, as demonstrated in the current study. To account for these results, we developed a simple extension to the previous model: Mice assess the value of current choice options not only based on current sensory stimuli (perception-choice values) but also based on a memory of the previous trial's rewarded choice (memory-choice values). While the trial-by-trial updating of perception-choice values leads to choice repetition, the concurrent learning of memory-choice values can attenuate or increase the tendency to repeat, allowing for a more flexible weighing of the previous trial. This minimal extension to the previous model explains several surprising patterns in our data. First, mice only adapt the influence of the previous choice across different temporal regularities, while similarly repeating choices of temporally more distant trials. Second, in contrast to the fast timescale of choice history biases, swiftly decaying over the past seven trials, the adaptation of the 1-back choice weight to temporal regularities is slow, developing over hundreds of trials. Third, when acting on random stimulus sequences mice more strongly repeat the 2-back compared to 1-back choice. Strikingly, all three empirical observations are captured by a model with a fixed set of parameters governing trial-by-trial learning in environments with different temporal regularities. This suggests that the empirical patterns arise from an interaction of a fixed set of learning rules with the temporal structure of the stimulus sequences.

Past perceptual decision-making studies in humans have shown similar reduced or muted 1- relative to 2-back choice weights for random stimulus sequences[10,60]. Moreover, similar reduced 1- relative to 2-back choice weights have been observed outside the perceptual domain in the context of a competitive matching pennies game in monkeys[61]. Similar to perceptual decision-making about random stimulus sequences, the optimal strategy in the matching pennies game is to make history-independent decisions. Instead, monkeys tend to repeat past decisions of their opponent - a pattern that can be exploited to their disadvantage and which the authors explained with a reinforcement learning model. Intriguingly, however, monkeys appear to be able to downregulate the repetition tendency of the 1-back

choice specifically, thereby becoming less exploitable—a phenomenon that can be readily accounted for by the multi-trial reinforcement learning model. Together, these findings suggest that similar multi-trial learning strategies might hold across decision-making contexts and species.

At the neural level, we found that dopamine release in the DLS closely tracks behavioral biases and reward predictions of the multi-trial reinforcement learning model. The activity of midbrain dopamine neurons is thought to play a pivotal role in learning from past rewards, encoding predicted value prior to outcome, and reward prediction error after outcome[24,62]. Similarly, during perceptual decisions, dopamine signals encode predicted values and reward prediction errors graded by both reward value and sensory confidence and are causally involved in learning from past perceptual decisions[21,52–55,63,64]. In consonance, we found that dopamine release in the DLS is positively scaled with the current stimulus contrast during the stimulus period, in line with signaling predicted value, but negatively scaled during reward processing, in line with encoding a reward prediction error. Trial-by-trial changes in these dopamine signals closely tracked behavioral biases and reward predictions of the multi-trial reinforcement learning model. Our finding that dopamine release not only reports a perceptual prediction, but also memory-based predictions is in line with past research indicating that midbrain dopamine neurons are sensitive to contextual information signaled by trial history[65]. Importantly, dopaminergic pathways in the dorsal striatum have been proposed to be involved in choice selection[66–68], and transient stimulation of dorsal striatal D1 neurons mimicked an additive change in choice values during decision-making[69]. Therefore, the history-dependent dopamine release in the DLS might be directly involved in promoting the adaptive behavioral choice history biases observed in the current study. Future studies that causally manipulate striatal dopamine release will be necessary to test this hypothesis.

We demonstrate that crucial signatures of choice history biases observed in the current study generalize across datasets, such as those by the International Brain Laboratory[19,20] (IBL), which uses a similar experimental setup. However, our manipulation of temporal regularities diverges from block switches used in the additional experimental manipulations of the IBL in important ways. While the full task of the IBL involves blockwise (i.e., average of 50 trials) manipulations of stimulus priors, the current study manipulates the local transition probability between successive trials, while keeping longer-term stimulus statistics balanced, therefore presenting a more subtle manipulation of input statistics. In particular, the use of alternating stimulus sequences enabled us to test whether mice learn to alternate from a previously rewarded choice, demonstrating that mice exhibit a flexible dependence on the previous trial given the prevailing temporal regularity.

The current study, while providing important insights into behavioral, computational, and neural bases of choice history biases, is not without limitations. First, while the multi-trial reinforcement learning model provides a parsimonious account of how mice rely on past rewarded choices, it does not adequately capture choice biases following unrewarded (error) trials. In particular, mice exhibited a bias to repeat unrewarded choices with similar strength across 1- to 7-back trials, indicating a slowly fluctuating tendency to repeat the same unrewarded choice (Supplementary Fig. 1d). This tendency is not recapitulated by our model (Supplementary Fig. 15), and differs from human behavior, which is characterized by choice alternation after errors[60]. It likely reflects both session-by-session changes in history-independent response biases as well as periods of task disengagement in which mice ignore stimuli and instead repeatedly perform the same choice[31]. Indeed, we found that mice repeated the previous incorrect choice when they were disengaged, but tended to alternate after errors when engaged with the task (Supplementary Fig. 3). Thus, when

focusing our analyses on periods of high task engagement, mice treated past incorrect trials more similar to humans and more consistent with a reinforcement learning agent, which predicts choice alternation following an unrewarded trial. However, the low proportion of error trials and their heterogeneity complicate a straightforward assessment of post-error responses. Nevertheless, they will be an important subject of investigation in future experimental and theoretical work.

Second, our conclusions are likely limited to adaptive history biases in settings involving trial-by-trial feedback. The presence of feedback, common in animal research, enables observers to learn most from maximally uncertain events, which is crucial for explaining how low decision confidence leads to strong choice repetition biases observed in this and previous datasets[9,20,21,32]. However, choice history biases occur in a wide range of experimental paradigms, many of which do not provide trial-by-trial feedback[1,70]. In the absence of feedback, human observers are more likely to repeat a previous choice when it was associated with high rather than low decision confidence[10,16,70–72], opposite to the current and past findings, and consistent with Bayesian models of choice history biases[73–76]. Thus, there are multiple ways through which observers can leverage the past to facilitate future behavior, and the resulting perceptual choice history biases are likely subserved by a variety of different computations, such as learning[32] and inference[77]. As such, while our model offers an explanation for perceptual choice history biases and their dopaminergic signatures, it does not necessarily exclude other theoretical frameworks.

Third, since we found signatures of adaptation to trial history even in the neutral (random) environment (see Fig. 4a–c), we focused on measuring choice behavior and dopamine release using random stimulus sequences, maximizing the number of trials in this condition. Importantly, we discovered that the choice history kernel in the neutral environment exhibited an important diagnostic to distinguish between the single- and multi-trial reinforcement learning models, namely the increase in 1- to 2-back choice history weight, which was recapitulated by the dopamine data. Nevertheless, it would be interesting to record dopamine release also during repeating and alternating sessions, and to investigate whether dopamine tracks the slow adaptation to the statistics of the environment across hundreds of trials.

Our results demonstrate that mice can flexibly adapt their choice history biases to different regularities of stimulus sequences in a visual decision-making paradigm. We show that a simple model-free POMDP RL algorithm based on multi-trial belief states accounts for the observed adaptive history biases and that striatal dopamine release closely follows the reward predictions of this algorithm. Our results suggest that choice history biases arise from continual learning that enables animals to exploit the temporal structure of the world to facilitate successful behavior.

## Methods

### Animals

The data for all experiments were collected from a total of 17 male C57BL/6J mice from Charles River UK, aged 10–30 weeks. The data of the behavioral experiment manipulating temporal regularities were collected from 10 mice. Of these mice, 3 animals also completed the experiment investigating sensory adaptation. Furthermore, we conducted dopamine recordings during perceptual decision-making in 6 mice. One of these mice also completed the sensory adaptation experiment. One mouse participated only in the sensory adaptation experiment. Mice were kept on a 12 h dark/light cycle, with an ambient temperature of 20–24° Celsius, and 40% humidity. All experiments were conducted according to the UK Animals Scientific Procedures Act (1986) under appropriate project and personal licenses.

## Surgery

Mice were implanted with a custom metal head plate to enable head fixation. To this end, animals were anesthetized with isoflurane and kept on a heating pad. Hair overlying the skull was shaved and the skin and the muscles over the central part of the skull were removed. The skull was thoroughly washed with saline, followed by cleaning with a sterile cortex buffer. The head plate was attached to the bone posterior to bregma using dental cement (Super-Bond C&B; Sun Medical).

For dopamine recording experiments, after attaching the headplate, we made a craniotomy over the left or right DLS. We injected 460 nL of diluted viral construct (pAAV-hsyn-GRAB$_{DA2m}$) into the left or right DLS (AP: +0.5 mm from bregma; ML: ±2.5 mm from midline; DV: 2.8 mm from dura). We implanted an optical fiber (200 mm, Neurophotometrics Ltd) over the DLS, with the tip 0.3 mm above the injection site. The fiber was secured to the head plate and skull using dental cement.

## Materials and apparatus

Mice were trained on a standardized behavioral rig, consisting of an LCD screen (9.7" diagonal), a custom 3D-printed mouse holder, and a head bar fixation clamp to hold a mouse such that its forepaws rested on a steering wheel[19,20]. Silicone tubing controlled by a pinch valve was used to deliver water rewards to the mouse. The general structure of the rig was constructed from Thorlabs parts and was placed inside an acoustical cabinet. The experiments were controlled by freely available custom-made software[78], written in MATLAB (Mathworks). Data analyses were performed with custom-made software written in Matlab 2020b, R (version 3.6.3), and Python 3.7. The GLM-HMM analysis was performed with the openly available glmhmm package (https://github.com/irisstone/glmhmm).

## Visual decision-making task

Behavioral training in the visual decision-making task started at least 5 days after the surgery. Animals were handled and acclimatized to head fixation for at least 3 days, and then trained in a 2-alternative forced choice visual detection task[19]. After mice kept the wheel still for at least 0.7 to 0.8 s, a sinusoidal grating stimulus of varying contrast appeared on either the left or right side of the screen (±35° azimuth, 0° altitude). Grating stimuli had a fixed vertical orientation, were windowed by a Gaussian envelope (3.5° s.d.), and had a spatial frequency of 0.19 cycles/° with a random spatial phase. Concomitant to the appearance of the visual stimulus, a brief tone was played to indicate that the trial had started (0.1 s, 5 kHz). Mice were able to move the grating stimulus on the monitor by turning a wheel located beneath their forepaws. If mice correctly moved the stimulus 35° to the center of the screen, they immediately received a water reward (2–3 µL). Conversely, if mice incorrectly moved the stimulus 35° towards the periphery or failed to reach either threshold within 60 s a noise burst was played for 0.5 s and they received a timeout of 2 s. The inter-trial interval was randomly sampled from a uniform distribution between 0.5 and 1 s (1 and 3 s in the dopamine recording experiment). In the initial days of training, only 100% contrast stimuli were presented. Stimuli with lower contrasts were gradually introduced after mice exhibited sufficiently accurate performance on 100% contrast trials (>70% correct). During this training period, incorrect responses on easy trials (contrast ≥ 50%) were followed by "repeat" trials, in which the previous stimulus location was repeated. The full task included six contrast levels (100, 50, 25, 12.5, 6.25 and 0% contrast). Once mice reached stable behavior on the full task, repeat trials were switched off, and mice proceeded to the main experiment.

In the main experiment (Figs. 1–3), we investigated whether mice adapt their choice history biases to temporal regularities. To this end, we manipulated the transitional probabilities between successive stimulus locations (left or right) across experimental sessions.

Specifically, the probability of a repetition was defined as follows:

$$P(\text{stimulus repetition}) = 1 - P(\text{stimulus alternation})$$
$$= P(\text{stimulus}_n = \text{left}|\text{stimulus}_{n-1} = \text{left}) \quad (6)$$
$$= P(\text{stimulus}_n = \text{right}|\text{stimulus}_{n-1} = \text{right})$$

where $n$ indexes trials. The repetition probability was held constant within each session but varied across experimental sessions, which were run on different days. In the Neutral environment, the repetition probability was set to 0.5, yielding entirely random stimulus sequences. In the Repeating and Alternating environments, the repetition probability was set to 0.8 and 0.2, respectively. For eight out of ten mice, the order of environments was pseudo-randomized such that three consecutive Repeating or Alternating sessions were interleaved with two consecutive Neutral sessions. For the remaining two mice, the environments were presented in random order.

Experimental sessions in which mice showed a high level of disengagement from the task were excluded from further analysis, based on the following criteria. We fit a psychometric curve to each session's data, using a maximum likelihood procedure:

$$P(\text{"Right"}) = \gamma + (1 - \gamma - \lambda)F(c; \alpha, \beta) \quad (7)$$

where P("Right") describes the mouse's probability to give a rightward response, F is the logistic function, $c$ is the stimulus contrast, $\gamma$ and $\lambda$ denote the right and left lapse rates, $\alpha$ is the bias and $\beta$ is the contrast threshold. We excluded sessions in which the absolute bias was larger than 0.16, or either left or right lapse rates exceeded 0.2. We further excluded sessions in which the choice accuracy on easy 100% contrast trials was lower than 80%. This led to the exclusion of 56 out of 345 sessions (16%). Finally, we excluded trials in which the response time was longer than 12 s, thereby excluding 1507 out of 128,490 trials (1.2%).

## Probabilistic choice model

In order to quantify the mice's choice history biases across the three environments with different temporal regularities, we fitted a probabilistic choice model to the responses of each mouse. In particular, we modeled the probability of the mouse making a rightward choice as a weighted sum of the current trial's sensory evidence, the successful and unsuccessful response directions of the past seven trials, and a general bias term, passed through a logistic link function:

$$P(\text{"Right"}) = \frac{1}{1 + e^{-z}} \quad (8)$$

where $z$ is the decision variable, which is computed for each trial $i$ in the following way:

$$z(i) = \sum_c w_c I_c(i) + \sum_{n=1}^{7} w_n^+ r^+(i-n) + w_n^- r^-(i-n) + w_0 \quad (9)$$

$w_c$ is the coefficient associated with contrast $I_c$ and $I_c$ is an indicator function, which is 1 if contrast $c$ was presented on trial $i$ and 0 otherwise. Coefficients $w_n^+$ and $w_n^-$ weigh the influence of the correct (+) and incorrect (−) choices of the past seven trials, denoted by $r^+$ and $r^-$, respectively. Here, $r^+$ was −1 if the correct n-back choice was left, +1 if it was right, and zero if the n-back choice was incorrect. Likewise, $r^-$ was −1 if the incorrect n-back choice was left, +1 if it was right, and zero if the n-back choice was correct. $w_0$ is a constant representing the overall bias of the mouse. We chose to model a temporal horizon of the past seven trials, since the autocorrelation in the stimulus sequences introduced by the transition probabilities decayed over this timeframe and were negligible beyond seven trials back (see Supplementary

Fig. 1e). It is important to model choice history kernels that cover the timeframe of autocorrelations in the stimulus sequences, in order to avoid long-term history biases and long-term autocorrelations to confound the estimate of short-term choice history weights across the different environments. While it is possible that mice exhibit even more slowly fluctuating history biases, beyond seven trials back, such slow biases would not differentially bias choice weights across the different environments.

We fitted the probabilistic choice model separately to the response data of each mouse in each environment, using a lasso regression implemented in glmnet[79]. The regularization parameter $\lambda$ was determined using a 10-fold cross-validation procedure.

We further investigated how correct choice weights $w_n^+$ developed across consecutive days when transitioning from the neutral into a regular environment (neutral → repeating or neutral → alternating). To this end, we subdivided the data into regular (repeating/alternating) sessions, which were preceded by a neutral session (day 1), or preceded by a neutral followed by one or two regular sessions of the same kind (days 2 and 3). We fitted the probabilistic choice model to the choices of days 1, 2, and 3 using the same procedure described above.

Finally, we tested whether correct choice weights $w_n^+$ in the neutral environment depended on the temporal regularity that mice experienced in the preceding session. Hence, we fitted the probabilistic choice model to neutral session data, separately for sessions preceded by a repeating or alternating environment.

## Parameter recovery analysis

In order to investigate whether the probabilistic choice model was able to recover choice history kernels in the face of autocorrelated stimulus sequences, we conducted a parameter recovery analysis. First, we obtained a set of "ground truth" choice history kernels by fitting the probabilistic choice model to the observed choices of each mouse in each environment, as previously described. We then used these postulated ground truth parameters to simulate synthetic choice data in response to stimulus sequences of all three environments. The simulated choice data was not identical to the empirical choice data due to the probabilistic nature of the model. However, it was generated according to the same stimulus and choice history weights that we reported previously. We simulated 100 synthetic datasets for each mouse. We then asked whether we could recover the ground truth choice history kernels when subjecting the simulated choice data to our analysis pipeline. We found that ground truth history kernels were accurately recovered, regardless of which stimulus sequence was used to simulate choices (Supplementary Fig. 1g–i). That is, an artificial observer with a neutral choice history kernel was estimated to have a neutral choice history kernel regardless of the stimulus sequence to which it responded (Supplementary Fig. 1g). Furthermore, this neutral history kernel was distinctly different from the repeating and alternating history kernels (Supplementary Fig. 1h and i). This indicates that our model fitting procedure is able to accurately recover choice history kernels of the shape that we report in the main results.

## Hidden Markov Model analysis of engaged and disengaged trials

We identified engaged and disengaged trials using a modeling framework based on hidden Markov Models[31] (HMM). In particular, we fit a HMM with two states, and their state-specific Bernoulli Generalized Linear Models (GLMs) to our neutral environment task data. The GLMs consisted of a stimulus regressor and stimulus-independent bias term. We hypothesized to obtain one state with high stimulus weight, reflecting high engagement with the task, and one state with low stimulus weight, reflecting disengagement from the task. This was indeed borne out in the data (Supplementary Fig. 3a and b). We repeated the probabilistic choice model analysis described above separately for engaged and disengaged current and 1- to 7-back trials.

## Reanalysis of data by the International Brain Laboratory

When analyzing correct choice history weights in the neutral environment, we observed that mice were more strongly biased to repeat their response given in the 2-back trial compared to the more recent 1-back response. One might surmise that this increase in choice repetition from 1- to 2-back trials could be driven by the exposure of the mice to multiple transition probabilities in the current study. In order to test whether this phenomenon was indeed particular to the current experimental design, involving stimulus sequences with biased transition probabilities, we analyzed a large, publicly available dataset of mice performing a similar visual decision-making task, which had not experienced biased transition probabilities[20]. We selected sessions in which mice had mastered the task, but before they were exposed to sessions involving blocked manipulations of stimulus locations (full task of the IBL study). Mice had therefore only experienced random stimulus sequences. Using the same exclusion criteria described above, we analyzed data of 99 mice in 583 sessions, comprising 471,173 choices. To estimate choice history weights, we fitted the same probabilistic choice model as described above to the data of each mouse. The data analyzed in the current study is available here:

https://figshare.com/articles/dataset/A_standardized_and_reproducible_method_to_measure_decision-making_in_mice_Data/11636748

## Reinforcement learning models

In order to investigate the computational principles underlying history bias adaptation, we adopted and extended a previously proposed Reinforcement Learning (RL) model based on a partially observable Markov decision process (POMDP)[21,33]. We will first describe the previously proposed model, which we term single-trial POMPD RL model, as this model's belief state was solely based on the current trial's visual stimuli. We will then describe an extension to this model, which we term multi-trial POMPD RL model. In addition to the current visual stimuli, the belief state of the multi-trial POMPD RL model incorporates a memory of the previous rewarded choice.

**Single-trial POMPD RL model.** In our visual decision-making task, the state of the current trial (left or right) is uncertain and therefore only partially observable due to the presence of low contrast stimuli and sensory noise. The model assumes that the agent forms an internal estimate $\hat{s}$ of the true signed stimulus contrast $s$, which is normally distributed with constant variance around the true stimulus contrast: $p(\hat{s}|s) = \mathcal{N}(\hat{s}; s, \sigma^2)$. Following Bayesian principles, the agent's belief about the current state is not limited to the point estimate $\hat{s}$, but consists of a belief distribution over all possible values of $s$ given $\hat{s}$. The belief distribution is given by Bayes rule:

$$p(s|\hat{s}) = \frac{p(\hat{s}|s)p(s)}{p(\hat{s})} \tag{10}$$

We assume that the prior belief about $s$ is uniform, yielding a Gaussian belief distribution $p(s|\hat{s})$ with the same variance as the sensory noise distribution and mean $\hat{s}$: $p(s|\hat{s}) = \mathcal{N}(s; \hat{s}, \sigma^2)$. The agent's belief that the stimulus was presented on the right side of the monitor, $P_R = p(s > 0|\hat{s})$, is given by:

$$P_R = \int_0^\infty p(s|\hat{s})\, ds \tag{11}$$

The agent's belief that the stimulus was presented on the left side is given by $P_L = 1 - P_R$.

The agent combines this belief state $[P_L, P_R]$ based on the current stimulus with stored values for making left and right choices in left and right perceptual states, given by $q_{choice,state}$ in order to compute expected values for left and right choices:

$$Q_L = \sum_{i \in \{L,R\}} P_i \cdot q_{L,P_i} \text{ and } Q_R = \sum_{i \in \{L,R\}} P_i \cdot q_{R,P_i} \quad (12)$$

Where $Q_L$ and $Q_R$ denote the expected value for left and right choices, respectively.

The agent then uses these expected values together with a softmax decision to compute a probability of making a rightward choice, p("Rightward choice"):

$$p(\text{"Rightward choice"}) = \frac{e^{Q_R/T}}{e^{Q_L/T} + e^{Q_R/T}} \quad (13)$$

where $T$ denotes the softmax temperature, introducing decision noise. The agent then decides for one of the two choice options using a biased coin flip, based on p("Rightward choice").

Following the choice, the agent observes the associated outcome $r$, which is 1 if the agents chose correctly and zero otherwise. It then computes a prediction error $\delta$ by comparing the outcome to the expected value of the chosen option $Q_C$:

$$\delta = r - Q_C \text{ where } Q_C = \begin{cases} Q_L & \text{if choice} = L \\ Q_R & \text{if choice} = R \end{cases} \quad (14)$$

Given this prediction error, the agent updates the values associated with the chosen option $q_{C,P_L}$ and $q_{C,P_R}$ by weighing the prediction error with a learning rate $\alpha$ and the belief of having occupied the particular state:

$$q_{C,P_L} \leftarrow q_{C,P_L} + \alpha \cdot P_L \cdot \delta \quad (15)$$

$$q_{C,P_R} \leftarrow q_{C,P_R} + \alpha \cdot P_R \cdot \delta \quad (16)$$

We allowed the agent to have two distinct learning rates that are used when predictions errors $\delta$ are positive ($\alpha^+$) or negative ($\alpha^-$).

The single-trial POMPD RL model thus had four free parameters, consisting of sensory noise $\sigma^2$, decision noise $T$, as well as positive and negative learning rates ($\alpha^+$ and $\alpha^-$).

**Multi-trial POMPD RL model.** We extended the single-trial POMPD RL model, by augmenting the state representation with a memory of the rewarded choice of the previous trial. Therefore, in addition to $P_L$ and $P_R$ which describe the agent's belief about the current stimulus being left or right, the agent's state further comprised memories $M_L$ and $M_R$, which were computed as follows:

$$\begin{aligned} M_L &= \begin{cases} \lambda & \text{if prev. choice} = L \text{ and rewarded} \\ 0 & \text{otherwise} \end{cases} \text{ and} \\ M_R &= \begin{cases} \lambda & \text{if prev. choice} = R \text{ and rewarded} \\ 0 & \text{otherwise} \end{cases} \end{aligned} \quad (17)$$

The parameter $\lambda$ reflects the memory strength of the agent, i.e., how strongly the agent relies on the memory of the previous rewarded choice. $\lambda$ was bounded by zero (no knowledge of the previous rewarded choice) and 1 (perfect knowledge of the previous rewarded choice). Therefore, besides learning values of pairings between perceptual states and choices ($q_{choice,P}$), the multi-trial agent additionally learned values of pairings between memory states and choices ($q_{choice,M}$). The expected values of left and right choice options were

thus computed as:

$$\begin{aligned} Q_L &= \sum_{i \in \{L,R\}} P_i \cdot q_{L,P_i} + \sum_{j \in \{L,R\}} M_j \cdot q_{L,M_j} \text{ and } Q_R \\ &= \sum_{i \in \{L,R\}} P_i \cdot q_{R,P_i} + \sum_{j \in \{L,R\}} M_j \cdot q_{R,M_j} \end{aligned} \quad (18)$$

The expected value of the current choice was thus both influenced by immediately accessible perceptual information of the current trial, as well as memory information carried over from the previous trial.

The multi-trial POMPD RL model had five free parameters, consisting of sensory noise $\sigma^2$, decision noise $T$, positive and negative learning rates ($\alpha^+$ and $\alpha^-$), and memory strength $\lambda$. The same learning rate was used to update perception-choice and memory-choice values.

**Multi-trial POMDP RL model with extended memory.** The memory of the multi-trial POMDP RL model was limited to the previous rewarded choice. We further extended this model, such that the agent's memory was based on multiple past trials. In particular, memory states $M_L$ and $M_R$ were computed as an exponentially weighted sum of past rewarded left and right choices:

$$M_L = M_0 \sum_{i=0}^{t-1} w_i I_L(c_i) \text{ and } M_R = M_0 \sum_{i=0}^{t-1} w_i I_R(c_i) \quad (19)$$

where $I_L$ and $I_R$ denote indicator functions, evaluated to 1 when the $i^{th}$ choice was a rewarded left or right choice, respectively, and zero otherwise. $M_0$ was the initial memory strength for the 1-back trial, and weights $w_i$ implemented an exponential decay function

$$w_i = \exp(-(t - i - 1)/\tau) \quad (20)$$

The exponential decay function was defined over elapsed trials, where $t$ denotes the index of the current trial and $i$ denotes the index of a previous trial. Thus, $\tau$ denotes the number of elapsed trials beyond the 1-back trial after which the contribution of a past choice to memory has decreased to $1/e = 0.37$ of its initial value. Similar to the multi-trial POMDP, the agent learned values of pairings between these exponentially weighted memories of past rewarded choices and the current choice ($q_{choice,M}$).

The multi-trial POMDP RL model with extended memory had six free parameters, consisting of sensory noise, decision noise, positive and negative learning rates, initial memory strength $M_O$, and exponential decay time constant $\tau$.

**Fitting procedure and model comparison.** We fit the single- and multi-trial POMPD RL models to the joint data of the neutral, repeating, and alternating environments pooled across mice. In particular, we used the empirical coefficients of the probabilistic choice model (see above) fit to the pooled data in order to define a cost function based on summary statistics of the mice's behavior[80]. The cost function was defined as the sum of squared difference between empirical and model coefficients, comprising the current stimulus weights of each environment, the 1- to 7-back correct choice weights of each environment, as well as the 1- to 7-back correct choice weights of neutral sessions following repeating and alternating sessions:

$$\begin{aligned} cost = &\sum_{env \in \{N., Rep., Alt.\}} \sum_c \left(w_{c,env}^{mouse} - w_{c,env}^{model}\right)^2 + \sum_{env \in \{N., Rep., Alt.\}} \sum_{n=1}^{7} \left(w_{n,env}^{+mouse} - w_{n,env}^{+model}\right)^2 \\ &+ \sum_{n=1}^{7} \left(w_{n,post-rep}^{+mouse} - w_{n,post-rep}^{+model}\right)^2 + \sum_{n=1}^{7} \left(w_{n,post-alt}^{+mouse} - w_{n,post-alt}^{+model}\right)^2 \end{aligned}$$

$$(21)$$

We minimized the cost function using Bayesian adaptive direct search (BADS[81]). BADS alternates between a series of fast, local Bayesian optimization steps and a systematic, slower exploration of a mesh grid. When fitting the single-trial POMPD RL model, we constrained the parameter space as follows. The sensory noise parameter $\sigma^2$ was constrained to the interval [0.05, 0.5], decision noise to [0.01,1], and positive and negative learning rates to [0.01, 1]. When fitting the multi-trial POMPD RL model, we additionally constrained the memory strength parameter to lie between zero (no memory) and 1 (perfect memory). The single-trial model was thus a special case of the multi-trial model without memory ($\lambda = 0$). That is, the single-trial model was nested in the multi-trial model. We repeated the optimization process from 9 different starting points to confirm converging solutions. Starting points were arranged on a grid with low and high learning rates $\alpha^+ = \alpha^- = [0.2, 0.8]$, and weak and strong memory $\lambda = [0.05, 0.15]$. Sensory noise $\sigma^2$ and decision noise $T$ were initialized with 0.1 and 0.3, respectively. In addition to the 8 starting points spanned by this grid, we added a 9th starting point determined by manual optimization ($\sigma^2 = 0.14$, $\alpha^+ = 0.95$, $\alpha^- = 0.95$, $T = 0.34$, $\lambda = 0.05$). The best fitting parameters for the multi-trial model were: $\sigma^2 = 0.09$, $\alpha^+ = 0.96$, $\alpha^- = 0.97$, $T = 0.36$, $\lambda = 0.07$, and multiple starting points converged to similar solutions. The best fitting parameters for the single trial model were: $\sigma^2 = 0.1$, $\alpha^+ = 0.41$, $\alpha^- = 1$, $T = 0.4$. We formally compared the best-fitting single- and multi-trial models using an $F$-test for nested models, as well as the Bayesian information criterion (BIC).

## Visual decision-making task investigating spatially-specific adaptation biases

We trained mice ($n = 5$) in an alternative version of the visual decision-making task, in order to test whether the decreased tendency to repeat the 1- relative to 2-back choice and the increased probability to repeat decisions based on low sensory evidence could be due to spatially-specific sensory adaptation to stimulus contrast. The experimental design was similar to the standard task, with the important exception that grating stimuli were smaller (2° s.d. Gaussian envelope) and the vertical location of the stimuli was randomly varied between ±15° altitude (3 mice) or ±10° altitude (2 mice) across trials. We reasoned that the vertical distance of 20−30 visual degrees would be sufficient to stimulate partly non-overlapping visual cortical neural populations, given receptive field sizes of 5−12 visual degrees (half-width at half-maximum) in primary visual cortex[82]. We trained mice to report the horizontal location of the visual stimulus (left/right), independent of its vertical location (high/low). This allowed us to manipulate whether successive stimuli were presented at the same or different spatial location (lower and upper visual field), even when those stimuli required the same choice (left or right; Fig. 4e). We applied the same session and trial exclusion criteria as for the main task. To analyze the dependence of current choices on the previous trial, we selected trials that were preceded by a correctly identified stimulus on the same side as the current stimulus. We binned trials into those preceded by a low or high contrast stimulus (6.25 and 100%), presented at the same or different vertical location. For each of the four bins, we computed the probability that the mouse repeats the previous choice, averaged across current stimulus contrasts. Owing to the fact that current and previous stimuli were presented at the same side, this repetition probability was larger than 0.5, but could nevertheless be modulated by previous contrast and vertical location. We tested the effects of previous contrast (high/low) and vertical location (same/different) with a 2 × 2 repeated-measures ANOVA. Furthermore, to provide statistical evidence against the hypothesis of spatially-specific sensory adaptation, we conducted a Bayes Factor analysis, quantifying evidence for the one-sided hypothesis that, due to a release from sensory adaptation, mice would be more likely to repeat a previous choice when a previous high contrast stimulus was presented at a different

rather than same spatial location (Fig. 4g). The Bayes Factor was calculated with a default prior scale of 0.707.

## Dopamine recording experiment

To measure dopamine release in the DLS, we employed fiber photometry[83,84]. We used a single chronically implanted optical fiber to deliver excitation light and collect emitted fluorescence. We used multiple excitation wavelengths (470 and 415 nm), delivered on alternating frames (sampling rate of 40 Hz), serving as target and isosbestic control wavelengths, respectively. To remove movement and photobleaching artifacts, we subtracted the isosbestic control from the target signal. In particular, for each session, we computed a least-squares linear fit of the isosbestic to the target signal. We subtracted the fitted isosbestic from the target signal and normalized by the fitted isosbestic signal to compute ΔF/F:

$$\frac{\Delta F}{F} = \frac{\text{Signal}_{470} - \text{Fitted Control}_{415}}{\text{Fitted Control}_{415}} \quad (22)$$

The resulting signal was further high-pass filtered by subtracting a moving average (25 s averaging window) and z-scored. For the main analyses, we aligned the z-scored ΔF/F to stimulus or reward onset times and baselined the signal to the pre-stimulus period of the current trial (−0.5−0 s relative to stimulus onset). For assessing whether our results could be explained by a slow carryover of the previous trial's dopamine response, we conducted an alternative analysis, in which we baselined the current trial's dopamine signal to the previous trial's pre-stimulus period and assessed the dopamine signal before the onset of the current stimulus.

We trained mice ($n = 6$) in the same visual decision-making task as described above, with the exception that we increased the inter-trial delay period (ITI), sampling from a uniform distribution between 1 and 3 s to allow the dopamine signal to return to baseline before the next trial. Due to the increased ITI, the median inter-stimulus interval was 5.55 s. To maximize the number of trials, we only presented neutral (random) stimulus sequences.

## Reporting summary

Further information on research design is available in the Nature Portfolio Reporting Summary linked to this article.

## Data availability

The behavioral and photometry data generated in this study have been deposited in the Figshare database under accession code https://doi.org/10.6084/m9.figshare.24179829. The behavioral data of the International Brain Laboratory used in this study are available in the Figshare database under accession code https://doi.org/10.6084/m9.figshare.11636748.v7. Source data are provided with this paper.

## Code availability

The code generated in this study has been deposited in the Figshare database under accession code https://doi.org/10.6084/m9.figshare.24179829.

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

## Acknowledgements

This research was supported by the following grants: NWO Rubicon Fellowship (019.211EN.006) and HFSP Long-Term Fellowship (LT0045/2022-L) to M.F.; BBSRC (BB/S006338/1) and MRC (MC_UU_00003/1) to R.B.; Sir Henry Dale Fellowship from the Wellcome Trust and Royal Society (213465/Z/18/Z) to A.L. For the purpose of Open Access, the author has applied a CC BY public copyright licence to any Author Accepted Manuscript version arising from this submission.

## Author contributions

M.F. and A.L. conceived and designed the study. M.F., A.M., and L.S. performed the experiments and acquired the data. M.F. performed the formal analysis, with inputs from S.L.G., R.B., and A.L. M.F. and A.L. interpreted the results, with contributions from S.L.G. and R.B. M.F. and A.L. wrote the manuscript, with valuable revisions from all authors.

## Competing interests

The authors declare no competing interests.
