## [Peer Review File · Nature Communications]

Temporal regularities shape perceptual decisions and striatal dopamine signalsEditorial Note: This manuscript has been previously reviewed at another journal that is not operating a transparent peer review scheme. This document only contains reviewer comments and rebuttal letters for versions considered at *Nature Communications*.

REVIEWERS' COMMENTS

Reviewer #1 (Remarks to the Author):

The questions raised in my initial review have been addressed by the authors in very satisfying ways. Overall, the complementary analyses are well-executed and enhance the relevance of the article. In particular, the analysis on task engagement & disengagement states is an outstanding addition to the original manuscript. These changes have been seamlessly integrated into the main text. In my opinion, the current submission represents a major improvement over the original and is ready for publication. I have added a few minor comments that would benefit from being addressed for final publication. I would like to thank the authors for the clarity and quality of their responses.

1. Model recovery analysis for the probabilistic choice model

The two-pronged approach adopted by the authors to support the claim of their main behavioral finding - this adaptive 1-back choice history bias across environments - is convincing. The parameter recovery analysis is well-done and their conclusions unambiguously support their claim. The model comparison analysis is interesting, though it is intriguing that model 2 is significantly better than model 1, yet model 3, which is just model 2 plus additional parameters, is not significantly better than model 1. Does the cross-validated binomial deviance measure take into account the number of parameters of the models?

2. Incorrect choice weight values and comparison with a simpler model

I am pleased to see that my questions about the incorrect trials choice history weights have led the authors to address the question of engagement and disengagement in the

task, which in my opinion is an interesting addition to the manuscript. It is very pertinent to use the Ashwood framework to identify the disengaged trials in the task. The behavioral differences found between these 2 states are very compelling and supportive of other claims made in the manuscript. It is great to see that mice in engaged states do exhibit a tendency to alternate from their previous incorrect choice!

I thank the authors for conducting the suggested analysis by fitting a model with choice weight unconditioned on the past trials success to the neutral data of the task. It seemed like an important control to have, even though the inclusion of accuracy metrics in the main text makes it clear that the unconditioned choice weights are going to be similar to the correct choice weight due the low proportion of incorrect trials. It is also great to see the IBL findings replicated in the block-biased environment. The positive incorrect 1-back choice weight found on the IBL data suggests that there must be a significant number of disengaged trials in the IBL dataset as well.

3. Implementation of memory in the Multi-trial belief state model

The authors investigated how a multi-trial belief model with a more principled memory trace would perform in their experiment. From the figure R5.d, I see that the choice history kernel for $\tau=1$ is closer to behavior than that of the 1-back memory model, notably that its 1-back choice weight is smaller than the 2-back choice weight, which was not obvious from the model's definition. It would be interesting to see this figure across the three environments like in figures 3.b & 3.f instead of just in the neutral environment. It would also be beneficial to plot the lines for $\tau=0.17$ on the same plot as it represents the best fit.

4. No performance comparison

The authors addressed the behavioral performances of mice and models in more detail. It was important to address the accuracy of both models to clearly show that the single-belief trial was not able to reproduce the performances of the subjects in an alternating environment. I understand that, both as a candidate model in literature for explaining

choice history bias, and as a baseline for the multi-trial model, the inclusion of the single-trial model in the paper is indeed justified.

5. Motivations behind the probabilistic choice model temporal horizon

The authors clarified the motivations behind this choice of temporal horizon. The reasoning behind covering at least the timeframe of autocorrelations in the stimulus sequence is clearly exposed, and the weight significance analysis is a valuable addition to it.

6. Issue with references used to support their finding on this type of choice history bias

The authors delved more into the two apparent differences between their results in mice and previous findings in humans. It is a compelling hypothesis that this discrepancy might be driven by the difference in the rate of adaptation to new temporal regularities, and it is nicely supported by the modeling.

7. Performance in the alternating environment

The authors included the overall accuracy statistics in the result section, which answers my question.

8. Individual-level relationship between 1-back & 2-back choice weights

The authors included a supplementary figure of the correct choice history kernels of individual mice. It is great to see that the effect is so clean at the individual level.

9. Incorrect choice weight values for the model

The authors clearly addressed the limitation of their RL model to capture the behavior of mice in response to unrewarded trials.

I'd like to point the authors to recent article which might be of great interest for the continuation of their work. It captures the suboptimal choice history biases of mice in a similar task via a low-rank RNN modeling approach, thus attributing these biases to a limited computational power : Recurrent networks endowed with structural priors explain suboptimal animal behavior, Molano-Mazon et al., 2023, Current Biology.

10. Use of the IBL dataset

There was indeed a misunderstanding and the authors clarified which sessions of the IBL dataset were analyzed. In addition, they also extended the use of their pipeline to block-task sessions of the IBL data. It's reassuring to see that the choice weights they found in the biased IBL sessions align with the results of the IBL collaboration (with this exponential decaying kernel over past choices).

Reviewer #2 (Remarks to the Author):

I commend the authors for this revision, which addresses my previous (anyway fairly presentational) comments. I continue to think the behavioral/modeling aspects of this study are extremely illuminating, clear, and unexpected while also being simple. The dopamine results are a nice additional confirmation, cleaned up a bit in this version and with an interesting speculation that may explain one way in which they seem to differ from the model's predictions. I have no further comments.

Reviewer #3 (Remarks to the Author):

The reviewers have mostly responded to my comments. I have no technical objections and the data seems high quality. I think that the addition of the GLM-HMM and expanded discussion of alternative explanations for this phenomenon adds to the manuscript.

In the end, I think that the finding that the mice adapt their choices to the temporal structure of the environment is interesting. However, I am still not convinced that the authors have demonstrated that DLS dopamine is meaningfully contributing to this behavioral phenomenon or providing evidence for their model:

1) It definitely helps to include the deltaQ predictions for the two models in figure 5 and I think the data presentation has improved here overall. However, although the similar shape of the dopamine response is suggestive, I'm not sure how meaningful the comparison between 1 and 2 trials back is in 5i given that neither of these points are statistically distinguishable from 0 (if I'm interpreting fig 5i correctly--it's hard to tell what n.s. is referring to in that figure and it's not explained in the legend). In other words, I don't think it's meaningful to say there's a bigger difference in dopamine release between alternation and repetition relative to 2 trials back vs. 1 trial back if dopamine isn't distinguishing between alternation and repetition in the first place. This may be a power issue given that the amplitude of these differences is so small.

2) The authors are arguing that dopamine release reflects the overall Q value (inclusive of the perceptual belief state and the memory term). Further, I assume the authors think there's some feature of dopamine activity that is specific to the memory term, the value of which is lower for stimulus repetition than alternation trials (similar to the dopamine response). I think the hypothesis that the memory term weights more on the dopamine activity than the perceptual belief state is an interesting one, but why not explicitly test this. If the prediction is that dopamine reflects expected value, since the authors are estimating expected value and its hypothesized components, a more direct analysis is to correlate dopamine release with value estimated by their model (with a regression for example to model how measured dopamine is modulated by trial-by-trial stimulus contrast, $Q_{\text{perception}}$ and Q_{memory}). This would be more direct than showing that the relative position of 1 vs. 2 trial back dopamine release on repetition-alternation trials is the same as the relative position of Q on repetition-alternation trials (particularly given point 1 where it doesn't even seem like dopamine release is significantly different on repetition and alternation trials).

Minor point:

1) Looking back at the methods, I couldn't find how $q_{L,M}$ and $q_{R,M}$ are updated. Is the learning rate the same for the memory value updates and the perceptual value updates?

2) Sometimes repeat/repetition and alternate/alternation are used to describe the choice of the mouse (e.g. logistic regression analyses) and sometimes to refer to the stimulus (e.g., dopamine analyses). I found this to be difficult...maybe preceding with choice or stimulus (i.e., choice repetition, stimulus repetition) would help readability.

Reviewer #1 (Remarks to the Author):

The questions raised in my initial review have been addressed by the authors in very satisfying ways. Overall, the complementary analyses are well-executed and enhance the relevance of the article. In particular, the analysis on task engagement & disengagement states is an outstanding addition to the original manuscript. These changes have been seamlessly integrated into the main text. In my opinion, the current submission represents a major improvement over the original and is ready for publication. I have added a few minor comments that would benefit from being addressed for final publication. I would like to thank the authors for the clarity and quality of their responses.

We would like to thank the reviewer for the positive and constructive comments. We are pleased to hear that they find the revised manuscript a major improvement over the original.

1. Model recovery analysis for the probabilistic choice model

The two-pronged approach adopted by the authors to support the claim of their main behavioral finding - this adaptive 1-back choice history bias across environments - is convincing. The parameter recovery analysis is well-done and their conclusions unambiguously support their claim. The model comparison analysis is interesting, though it is intriguing that model 2 is significantly better than model 1, yet model 3, which is just model 2 plus additional parameters, is not significantly better than model 1. Does the cross-validated binomial deviance measure take into account the number of parameters of the models?

We surmise that Model 3 is not significantly better than Model 1 due to overfitting. The cross-validation procedure ensures that more complex models cannot benefit from fitting noise, which does not generalize from training to testing data. In some instances, overfitting on the training data can worsen performance on testing data, compared to a simpler model, and this is likely the case here. We agree with the reviewer that the parameter recovery analysis provides the clearest evidence for our conclusion, and therefore we focused on this analysis in the manuscript.

2. Incorrect choice weight values and comparison with a simpler model

I am pleased to see that my questions about the incorrect trials choice history weights have led the authors to address the question of engagement and disengagement in the task, which in my opinion is an interesting addition to the manuscript. It is very pertinent to use the Ashwood framework to identify the disengaged trials in the task. The behavioral differences found between these 2 states are very compelling and supportive of other claims made in the manuscript. It is great to see that mice in engaged states do exhibit a tendency to alternate from their previous incorrect choice!

I thank the authors for conducting the suggested analysis by fitting a model with choice weight unconditioned on the past trials success to the neutral data of the task. It seemed like an important control to have, even though the inclusion of accuracy metrics in the main text makes it clear that the unconditioned choice weights are going to be similar to the correct choice weight due to the low proportion of incorrect trials. It is also great to see the IBL findings replicated in the block-biased environment. The positive incorrect 1-back choice weight found on the IBL data suggests that there must be a significant number of disengaged trials in the IBL dataset as well.

We thank the reviewer for their comments. We agree that the distinction between engaged and disengaged trials, and the comparison to the IBL blocked-task data added valuable insights and increased the clarity of the manuscript.

3. Implementation of memory in the Multi-trial belief state model

The authors investigated how a multi-trial belief model with a more principled memory trace would perform in their experiment. From the figure R5.d, I see that the choice history kernel for $\tau=1$ is closer to behavior than that of the 1-back memory model, notably that its 1-back choice weight is smaller than the 2-back choice weight, which was not obvious from the model's definition. It would be interesting to see this figure across the three environments like in figures 3.b & 3.f instead of just in the neutral environment. It would also be beneficial to plot the lines for $\tau=0.17$ on the same plot as it represents the best fit.

We would like to clarify that the choice history kernel of the 1-back memory model provides the closest fit to the mice's empirical choice history kernel. To make this more clear in the manuscript, we followed the reviewer's advice and added the choice history kernel for a model with $\tau = 0.17$ (effectively the 1-back model) to Supplementary Fig. 7d.

4. No performance comparison

The authors addressed the behavioral performances of mice and models in more detail. It was important to address the accuracy of both models to clearly show that the single-belief trial was not able to reproduce the performances of the subjects in an alternating environment. I understand that, both as a candidate model in literature for explaining choice history bias, and as a baseline for the multi-trial model, the inclusion of the single-trial model in the paper is indeed justified.

Thank you.

5. Motivations behind the probabilistic choice model temporal horizon

The authors clarified the motivations behind this choice of temporal horizon. The reasoning behind covering at least the timeframe of autocorrelations in the stimulus sequence is clearly exposed, and the weight significance analysis is a valuable addition to it.

Thank you. We agree that it is a valuable addition.

6. Issue with references used to support their finding on this type of choice history bias

The authors delved more into the two apparent differences between their results in mice and previous findings in humans. It is a compelling hypothesis that this discrepancy might be driven by the difference in the rate of adaptation to new temporal regularities, and it is nicely supported by the modeling.

Thank you.

7. Performance in the alternating environment

The authors included the overall accuracy statistics in the result section, which answers my question.

Thank you.

8. Individual-level relationship between 1-back & 2-back choice weights

The authors included a supplementary figure of the correct choice history kernels of individual mice. It is great to see that the effect is so clean at the individual level.

Thank you.

9. Incorrect choice weight values for the model

The authors clearly addressed the limitation of their RL model to capture the behavior of mice in response to unrewarded trials.

I'd like to point the authors to a recent article which might be of great interest for the continuation of their work. It captures the suboptimal choice history biases of mice in a similar task via a low-rank RNN modeling approach, thus attributing these biases to a limited computational power : Recurrent networks endowed with structural priors explain suboptimal animal behavior, Molano-Mazon et al., 2023, Current Biology.

We thank the reviewer for pointing out this paper. We have cited the empirical paper on which this computational study is based (Hermoso-Mendizabal et al, 2020).

10. Use of the IBL dataset

There was indeed a misunderstanding and the authors clarified which sessions of the IBL dataset were analyzed. In addition, they also extended the use of their pipeline to block-task sessions of the IBL data. It's reassuring to see that the choice weights they found in the biased IBL sessions align with the results of the IBL collaboration (with this exponential decaying kernel over past choices).

We are glad that this has been clarified.

Reviewer #2 (Remarks to the Author):

I commend the authors for this revision, which addresses my previous (anyway fairly presentational) comments. I continue to think the behavioral/modeling aspects of this study are extremely illuminating, clear, and unexpected while also being simple. The dopamine results are a nice additional confirmation, cleaned up a bit in this version and with an interesting speculation that may explain one way in which they seem to differ from the model's predictions. I have no further comments.

We thank the reviewer for their positive assessment of our work.

Reviewer #3 (Remarks to the Author):

The reviewers have mostly responded to my comments. I have no technical objections and the data seems high quality. I think that the addition of the GLM-HMM and expanded discussion of alternative explanations for this phenomenon adds to the manuscript. In the end, I think that the finding that the mice adapt their choices to the temporal structure of the environment is interesting.

We are pleased that the reviewer appreciates our revisions and finds the behavioral findings interesting.

However, I am still not convinced that the authors have demonstrated that DLS dopamine is meaningfully contributing to this behavioral phenomenon or providing evidence for their model:

1) It definitely helps to include the deltaQ predictions for the two models in figure 5 and I think the data presentation has improved here overall. However, although the similar shape of the dopamine response is suggestive, I'm not sure how meaningful the comparison between 1 and 2 trials back is in 5i given that neither of these points are statistically distinguishable from 0 (if I'm interpreting fig 5i

correctly--it's hard to tell what n.s. is referring to in that figure and it's not explained in the legend). In other words, I don't think it's meaningful to say there's a bigger difference in dopamine release between alternation and repetition relative to 2 trials back vs. 1 trial back if dopamine isn't distinguishing between alternation and repetition in the first place. This may be a power issue given that the amplitude of these differences is so small.

The reviewer is interpreting Fig. 5i correctly. We have now clarified that Δ DAs of 1- and 2-back trials are not statistically different from zero in the figure legend. There appears to be some variability in overall DA release for repetitions and alternations across animals, unrelated to the n-back position, possibly due to other factors of DA signaling that are not captured by our model. This variability renders 1- and 2-back Δ DAs not significant. However, despite this overall variability, a within-subject comparison of 1- and 2-back Δ DA reveals a consistent increase in Δ DA from 1- to 2-back trials ($p = 0.017$), which is the specific prediction of the multi-trial RL model. Given that this contrast between 1- and 2-back Δ DA provides a sensitive comparison by removing overall variability across animals, and distinguishes between single- and multi-trial models, we believe that it constitutes a meaningful comparison.

2) The authors are arguing that dopamine release reflects the overall Q value (inclusive of the perceptual belief state and the memory term). Further, I assume the authors think there's some feature of dopamine activity that is specific to the memory term, the value of which is lower for stimulus repetition than alternation trials (similar to the dopamine response). I think the hypothesis that the memory term weights more on the dopamine activity than the perceptual belief state is an interesting one, but why not explicitly test this. If the prediction is that dopamine reflects expected value, since the authors are estimating expected value and its hypothesized components, a more direct analysis is to correlate dopamine release with value estimated by their model (with a regression for example to model how measured dopamine is modulated by trial-by-trial stimulus contrast, $Q_{\text{perception}}$ and Q_{memory}). This would be more direct than showing that the relative position of 1 vs. 2 trial back dopamine release on repetition-alternation trials is the same as the relative position of Q on repetition-alternation trials (particularly given point 1 where it doesn't even seem like dopamine release is significantly different on repetition and alternation trials).

We performed the analysis suggested by the reviewer. In particular we regressed trial-by-trial dopamine release (averaged between 0.2 and 0.5s relative to stimulus onset) onto the model-derived $Q_{C,P}$ and $Q_{C,M}$ - the perception-based and memory-based reward expectations for the chosen option. Note that $Q_{C,P}$ partly depends on the current stimulus contrast, which is why we did not include trial-by-trial stimulus contrast as an additional predictor. The linear regression model yielded positive coefficients for both $Q_{C,P}$ and $Q_{C,M}$ (0.92 and 0.86), which were both statistically greater than zero ($p < 0.001$ and $p = 0.018$, one-sided permutation test, shuffling trial-by-trial dopamine measurements within each session). This indicates that DLS dopamine release encodes both perception- and memory-based reward expectations, in agreement with our previous conclusions. However, it does not provide insights into how dopamine is influenced by the recent trial history and how the history-dependent modulation of dopamine distinguishes between single- and multi-trial reinforcement learning models, which are core questions of the current study. To address these questions, we devised an analysis contrasting stimulus repetitions and alternations, which reveals the similarity between the mice's and model's history kernels and the dopamine history kernel (see main manuscript). As this analysis provides a more direct insight into the relationship between dopamine and choice history biases, we would like to maintain this analysis approach.

Minor point:

1) Looking back at the methods, I couldn't find how $q_{L,M}$ and $q_{R,M}$ are updated. Is the learning rate the same for the memory value updates and the perceptual value updates?

The learning rate is indeed the same for perceptual and memory value updates. We have clarified this in the main text (p. 5, line 209) and Methods section (p. 15, line 791).

2) Sometimes repeat/repetition and alternate/alternation are used to describe the choice of the mouse (e.g. logistic regression analyses) and sometimes to refer to the stimulus (e.g., dopamine analyses). I found this to be difficult...maybe preceding with choice or stimulus (i.e., choice repetition, stimulus repetition) would help readability.

We thank the reviewer for this suggestion. When writing about the dopamine analysis, we now use the term *stimulus* repetition to avoid confusion with choice repetition biases.